# Model selection for contextual bandits

**Dylan J. Foster**
Massachusetts Institute of Technology
dylanf@mit.edu

**Akshay Krishnamurthy**
Microsoft Research NYC
akshay@cs.umass.edu

**Haipeng Luo**
University of Southern California
haipengl@usc.edu

## Abstract

We introduce the problem of model selection for contextual bandits, where a learner must adapt to the complexity of the optimal policy while balancing exploration and exploitation. Our main result is a new model selection guarantee for linear contextual bandits. We work in the stochastic realizable setting with a sequence of nested linear policy classes of dimension $d_1 < d_2 < \ldots$, where the $m^\star$-th class contains the optimal policy, and we design an algorithm that achieves $\tilde{O}(T^{2/3}d_{m^\star}^{1/3})$ regret *with no prior knowledge* of the optimal dimension $d_{m^\star}$. The algorithm also achieves regret $\tilde{O}\left(T^{3/4} + \sqrt{Td_{m^\star}}\right)$, which is optimal for $d_{m^\star} \geq \sqrt{T}$. This is the first model selection result for contextual bandits with non-vacuous regret for all values of $d_{m^\star}$, and to the best of our knowledge is the first positive result of this type for any online learning setting with partial information. The core of the algorithm is a new estimator for the gap in the best loss achievable by two linear policy classes, which we show admits a convergence rate faster than the rate required to learn the parameters for either class.

## 1 Introduction

Model selection is the fundamental statistical task of choosing a hypothesis class using data. The choice of hypothesis class modulates a tradeoff between approximation error and estimation error, as a small class can be learned with less data, but may have worse asymptotic performance than a richer class. In the classical statistical learning setting, model selection algorithms provide the following luckiness guarantee: If the class of models decomposes as a nested sequence $\mathcal{F}_1 \subset \mathcal{F}_2 \subset \cdots \mathcal{F}_m \subset \mathcal{F}$, the sample complexity of the algorithm scales with the statistical complexity of the smallest subclass $\mathcal{F}_{m^\star}$ containing the true model, even though $m^\star$ is not known in advance. Such guarantees date back to Vapnik's structural risk minimization principle and are by now well-known (Vapnik, 1982, 1992; Devroye et al., 1996; Birgé and Massart, 1998; Shawe-Taylor et al., 1998; Lugosi and Nobel, 1999; Koltchinskii, 2001; Bartlett et al., 2002; Massart, 2007). In practice, one may use cross-validation—the de-facto model selection procedure—to decide whether to use, for example, a linear model, a decision tree, or a neural network. That cross-validation appears in virtually every machine learning pipeline highlights the necessity of model selection for successful ML deployments.

We investigate model selection in contextual bandits, a simple interactive learning setting. Our main question is: *Can model selection guarantees be achieved in contextual bandit learning, where a learner must balance exploration and exploitation to make decisions online?*

Contextual bandit learning is more challenging than statistical learning on two fronts: First, decisions must be made online without seeing the entire dataset, and second, the learner's actions influence what data is observed ("bandit feedback"). Between these extremes is full-information online learning, where the learner does not have to deal with bandit feedback, but still makes decisions online. Even in this simpler setting, model selection is challenging, since the learner must attempt to identify the appropriate model class while making irrevocable decisions and incurring regret. Nevertheless, several prior works on so-called parameter-free online learning (McMahan and Abernethy, 2013;

Orabona, 2014; Koolen and Van Erven, 2015; Luo and Schapire, 2015; Foster et al., 2017; Cutkosky and Boahen, 2017) provide algorithms for online model selection with guarantees analogous to those in statistical learning. With bandit feedback, however, the learner must carefully balance exploration and exploitation, which presents a substantial challenge for model selection. At an intuitive level, the reason is that different hypothesis classes require different amounts of exploration, but either over- or under-exploring can incur significant regret (A detailed discussion requires a formal setup and is deferred to Section 2). At this point, it suffices to say that prior to this work, we are not aware of any adequate model selection guarantee that adapts results from statistical learning to any online learning setting with partial information.

We provide a new model selection guarantee for the linear stochastic contextual bandit setting (Chu et al., 2011; Abbasi-Yadkori et al., 2011). We consider a sequence of feature maps into $d_1 < d_2 < \ldots < d_M$ dimensions and assume that the losses are linearly related to the contexts via the $m^\star$-th feature map, so that the optimal policy is a $d_{m^\star}$-dimensional linear policy. We design an algorithm that achieves $\tilde{O}(T^{2/3} d_{m^\star}^{1/3})$ regret to this optimal policy over $T$ rounds, *with no prior knowledge of $d_{m^\star}$*. As this bound has no dependence on the maximum dimensionality $d_M$, we say that the algorithm adapts to the complexity of the optimal policy. All prior approaches suffer linear regret for non-trivial values of $d_{m^\star}$, whereas the regret of our algorithm is sublinear whenever $d_{m^\star}$ is such that the problem is learnable. Our algorithm can also be tuned to achieve $\tilde{O}\left(T^{3/4} + \sqrt{T d_{m^\star}}\right)$ regret, which matches the optimal rate when $d_{m^\star} \geq \sqrt{T}$.

At a technical level, we design a sequential test to determine whether the optimal square loss for a large linear class is substantially better than that of a smaller linear class. We show that this test has *sublinear* sample complexity: while learning a near-optimal predictor in $d$ dimensions requires at least $\Omega(d)$ labeled examples, we can estimate the improvement in *value* of the optimal loss using only $O(\sqrt{d})$ examples, analogous to so-called variance estimation results in statistics (Dicker, 2014; Kong and Valiant, 2018). Crucially, this implies that we can test whether or not to use the larger class without over-exploring for the smaller class.

## 2 Preliminaries

We work in the stochastic contextual bandit setting (Langford and Zhang, 2008; Beygelzimer et al., 2011; Agarwal et al., 2014). The setting is defined by a context space $\mathcal{X}$, a finite action space $\mathcal{A} \coloneqq \{1, \ldots, K\}$ and a distribution $\mathcal{D}$ supported over $(x, \ell)$ pairs, where $x \in \mathcal{X}$ and $\ell \in \mathbb{R}^{\mathcal{A}}$ is a loss vector. The learner interacts with nature for $T$ rounds, where in round $t$: (1) nature samples $(x_t, \ell_t) \sim \mathcal{D}$, (2) the learner observes $x_t$ and chooses action $a_t$, (3) the learner observes $\ell_t(a_t)$. The goal of the learner is to choose actions to minimize the cumulative loss.

Following several prior works (Chu et al., 2011; Abbasi-Yadkori et al., 2011; Agarwal et al., 2012; Russo and Van Roy, 2013; Li et al., 2017), we study a variant of the contextual bandit setting where the learner has access to a class of regression functions $\mathcal{F} : \mathcal{X} \times \mathcal{A} \to \mathbb{R}$ containing the Bayes optimal regressor

$$f^\star(x, a) \coloneqq \mathbb{E}\left[\ell(a) \mid x\right] \quad \forall x, a. \tag{1}$$

We refer to this assumption ($f^\star \in \mathcal{F}$) as *realizability*. For each regression function $f$ we define the induced policy $\pi_f(x) \coloneqq \operatorname{argmin}_a f(x, a)$. Note that $\pi^\star \coloneqq \pi_{f^\star}$ is the globally optimal policy, and chooses the best action on every context. We measure performance via regret to $\pi^\star$:

$$\operatorname{Reg} \coloneqq \sum_{t=1}^{T} \ell_t(a_t) - \sum_{t=1}^{T} \ell_t(\pi^\star(x_t)).$$

Low regret is tractable here due to the realizability assumption, and it is well known that the optimal regret is $\tilde{\Theta}\left(\sqrt{T \cdot \operatorname{comp}(\mathcal{F})}\right)$, where $\operatorname{comp}(\mathcal{F})$ measures the statistical complexity of $\mathcal{F}$. For example, $\operatorname{comp}(\mathcal{F}) = \log|\mathcal{F}|$ for finite classes, and $\operatorname{comp}(\mathcal{F}) = d$ for $d$-dimensional linear classes (Agarwal et al., 2012; Chu et al., 2011).[1]

**Model selection for contextual bandits.** We aim for refined guarantees that scale with the complexity of the optimal regressor $f^\star$ rather than the worst-case complexity of the class $\mathcal{F}$. To this end,

we assume that $\mathcal{F}$ is structured as a nested sequence of classes $\mathcal{F}_1 \subset \mathcal{F}_2 \subset \ldots \subset \mathcal{F}_M = \mathcal{F}$, and we define $m^\star := \min\{m : f^\star \in \mathcal{F}_m\}$. The model selection problem for contextual bandits asks:

> *Given that $m^\star$ is not known in advance, can we achieve regret scaling as $\tilde{O}(\sqrt{T \cdot \mathrm{comp}(\mathcal{F}_{m^\star})})$,*
> *rather than the less favorable $\tilde{O}(\sqrt{T \cdot \mathrm{comp}(\mathcal{F})})$?*

A slightly weaker model selection problem is to achieve $\tilde{O}\left(T^\alpha \cdot \mathrm{comp}(\mathcal{F}_{m^\star})^{1-\alpha}\right)$ for some $\alpha \in [1/2, 1)$, again without knowing $m^\star$. Crucially, the exponents on $T$ and $\mathrm{comp}(\mathcal{F}_{m^\star})$ sum to one, implying that we can achieve sublinear regret whenever $\mathrm{comp}(\mathcal{F}_{m^\star})$ is sublinear in $T$, which is precisely whenever the optimal model class is learnable. This implies that the bound, in spite of having worse dependence on $T$, adapts to the complexity of the optimal class with no prior knowledge.

We achieve this type of guarantee for linear contextual bandits. We assume that each regressor class $\mathcal{F}_m$ consists of linear functions of the form

$$\mathcal{F}_m := \left\{(x,a) \mapsto \langle \beta, \phi^m(x,a) \rangle \mid \beta \in \mathbb{R}^{d_m}\right\},$$

where $\phi^m : \mathcal{X} \times \mathcal{A} \to \mathbb{R}^{d_m}$ is a fixed feature map. To obtain a nested sequence of classes, and to ensure the complexity is monotonically increasing, we assume that $d_1 < d_2 < \ldots, d_M = d$ and that for each $m$, the feature map $\phi^m$ contains the map $\phi^{m-1}$ as its first $d_{m-1}$ coordinates.[2] If $m^\star$ is the smallest feature map that realizes the optimal regressor, we can write

$$f^\star(x,a) = \langle \beta^\star, \phi^{m^\star}(x,a) \rangle,$$

where $\beta^\star \in \mathbb{R}^{d_{m^\star}}$ is the optimal coefficient vector. In this setup, the optimal rate if $m^\star$ is known is $\tilde{O}(\sqrt{T d_{m^\star}})$, obtained by LinUCB (Chu et al., 2011).[3] Our main result achieves both $\tilde{O}(T^{2/3} d_{m^\star}^{1/3})$ regret (i.e., $\alpha = 2/3$) and $\tilde{O}\left(T^{3/4} + \sqrt{T d_{m^\star}}\right)$ regret without knowing $m^\star$ in advance.

**Related work.** The model selection guarantee we seek is straightforward for full information online learning and statistical learning. A simple strategy for the former setting is to use a low-regret online learner for each sub-class $\mathcal{F}_m$ and aggregate these base learners with a master Hedge instance (Freund and Schapire, 1997). Other strategies include parameter-free methods like AdaNormalHedge (Luo and Schapire, 2015) and Squint (Koolen and Van Erven, 2015). Unfortunately, none of these methods appear to adequately handle bandit feedback. For example, the regret bounds of parameter-free methods do not depend on the so-called "local norms", which are essential for achieving $\sqrt{T}$-regret in the bandit setting via the usual importance weighting approach (Auer et al., 2002). See Appendix B for further discussion.

In the bandit setting, two approaches we are aware of also fail: the Corral algorithm of Agarwal et al. (2017b), and an adaptive version of the classical $\epsilon$-greedy strategy (Langford and Zhang, 2008). Unfortunately, both algorithms require tuning parameters in terms of the unknown index $m^\star$, and naive tuning gives a guarantee of the form $\tilde{O}(T^\alpha \mathrm{comp}(\mathcal{F}_{m^\star})^\beta)$ where $\alpha + \beta > 1$. For example, for finite classes Corral gives regret $\sqrt{T} \log|\mathcal{F}_{m^\star}|$. This guarantee is quite weak, since it is vacuous when $\log|\mathcal{F}_{m^\star}| = \Theta(\sqrt{T})$ even though such a class admits sublinear regret if $m^\star$ were known in advance (see Appendix B). The conceptual takeaway from these examples is that model selection for contextual bandits appears to require new algorithmic ideas, even when we are satisfied with $O(T^\alpha \mathrm{comp}(\mathcal{F}_{m^\star})^{1-\alpha})$-type rates.

Several recent papers have developed adaptive guarantees for various contextual bandit settings. These include: (1) adaptivity to easy data, where the optimal policy achieves low loss (Allenberg et al., 2006; Agarwal et al., 2017a; Lykouris et al., 2018; Allen-Zhu et al., 2018), (2) adaptivity to smoothness in settings with continuous action spaces (Locatelli and Carpentier, 2018; Krishnamurthy et al., 2019), and (3) adaptivity in non-stationary environments, where distribution drift parameters are unknown (Luo et al., 2018; Cheung et al., 2019; Auer et al., 2018; Chen et al., 2019). The latter results can be cast as model selection with appropriate nested classes of *time-dependent* policies, but these results are incomparable to our own, since they are specialized to the non-stationary setting.

Interestingly, for multi-armed (non-contextual) bandits, several lower bounds demonstrate that model selection is *not* possible. The simplest of these results is Lattimore's pareto frontier (Lattimore, 2015), which states that for multi-armed bandits, if we want to ensure $O(\sqrt{T})$ regret against a single fixed arm instead of the usual $O(\sqrt{KT})$ rate, we must incur $\Omega(K\sqrt{T})$ regret to the remaining $K-1$ arms. This precludes a model selection guarantee of the form $\sqrt{T \cdot \text{comp}(\mathcal{A})}$ since for bandits, the statistical complexity is simply the number of arms. Related lower bounds are known for Lipschitz bandits (Locatelli and Carpentier, 2018; Krishnamurthy et al., 2019). Our results show that model selection *is* possible for contextual bandits, and thus highlight an important gap between the two settings.

In concurrent work, Chatterji et al. (2019) studied a similar model selection problem with two classes, where the first class consists of all $K$ constant policies and the second is a $d$-dimensional linear class. They obtain logarithmic regret to the first class and $O(\sqrt{Td})$ regret to the second, but their assumptions on the context distribution are strictly stronger than our own. A detailed discussion is deferred to the end of the section.

**Technical preliminaries and assumptions.**  For a matrix $A$, $A^\dagger$ denotes the pseudoinverse and $\|A\|_2$ denotes the spectral norm. $I_d$ denotes the identity matrix in $\mathbb{R}^{d \times d}$ and $\|\cdot\|_p$ denotes the $\ell_p$ norm. We use non-asymptotic big-$O$ notation, and use $\tilde{O}$ to hide terms logarithmic in $K$, $d_M$, $M$, and $T$.

For a real-valued random variable $z$, we use the following notation to indicate if $z$ is subgaussian or subexponential, following Vershynin (2012):

$$z \sim \mathsf{subG}(\sigma^2) \Leftrightarrow \sup_{p \geq 1}\{p^{-1/2}(\mathbb{E}|z|^p)^{1/p}\} \leq \sigma, \quad z \sim \mathsf{subE}(\lambda) \Leftrightarrow \sup_{p \geq 1}\{p^{-1}(\mathbb{E}|z|^p)^{1/p}\} \leq \lambda. \quad (2)$$

When $z$ is a random variable in $\mathbb{R}^d$, we write $z \sim \mathsf{subG}_d(\sigma^2)$ if $\langle \theta, z \rangle \sim \mathsf{subG}(\sigma^2)$ for all $\|\theta\|_2 = 1$ and $z \sim \mathsf{subE}_d(\lambda)$ if $\langle \theta, z \rangle \sim \mathsf{subE}(\lambda)$ for all $\|\theta\|_2 = 1$. These definitions are equivalent to many other familiar definitions for subgaussian/subexponential random variables; see Appendix C.1.

We assume that for each $m$ and $a \in \mathcal{A}$, $\phi^m(x,a) \sim \mathsf{subG}(\tau_m^2)$ under $x \sim \mathcal{D}$. Nestedness implies that $\tau_1 \leq \tau_2 \leq \ldots$, and we define $\tau = \tau_M$. We also assume that $\ell(a) - \mathbb{E}[\ell(a) \mid x] \sim \mathsf{subG}(\sigma^2)$ for all $x \in \mathcal{X}$ and $a \in \mathcal{A}$. Finally, we assume that $\|\beta^\star\| \leq B$. To keep notation clean, we use the convention that $\sigma \leq \tau$ and $B \leq 1$, which ensures that $\ell(a) \sim \mathsf{subG}(4\tau^2)$.

We require a lower bound on the eigenvalues of the second moment matrices for the feature vectors. For each $m$, define $\Sigma_m := \frac{1}{K} \sum_{a \in \mathcal{A}} \mathbb{E}_{x \sim \mathcal{D}}[\phi^m(x,a)\phi^m(x,a)^\top]$. We let $\gamma_m^2 := \lambda_{\min}(\Sigma_m)$, where $\lambda_{\min}(\cdot)$ denotes the smallest eigenvalue; nestedness implies $\gamma_1 \geq \gamma_2 \geq \ldots$. We assume $\gamma_m \geq \gamma > 0$ for all $m$, and our regret bounds scale inversely proportional to $\gamma$.

Note that prior linear contextual bandit algorithms (Chu et al., 2011; Abbasi-Yadkori et al., 2011) do not require lower bounds on the second moment matrices. As discussed earlier, the work of Chatterji et al. (2019) obtains stronger model selection guarantees in the case of two classes, but their result requires a lower bound on $\lambda_{\min}(\mathbb{E}[\phi(x,a)\phi(x,a)^\top])$ for all actions. Previous work suggests that advanced exploration is not needed under such assumptions (Bastani et al., 2017; Kannan et al., 2018; Raghavan et al., 2018), which considerably simplifies the problem.[4] As such, the result should be seen as complementary to our own. Whether model selection can be achieved without some type of eigenvalue condition is an important open question.

## 3 Model selection for linear contextual bandits

We now present our algorithm for model selection in linear contextual bandits, ModCB ("Model Selection for Contextual Bandits"). Pseudocode is displayed in Algorithm 1. The algorithm maintains an "active" policy class index $\widehat{m} \in [M]$, which it updates over the $T$ rounds starting from $\widehat{m} = 1$. The algorithm updates $\widehat{m}$ only when it can prove that $\widehat{m} \neq m^\star$, which is achieved through a statistical test called EstimateResidual (Algorithm 2). When $\widehat{m}$ is not being updated, the algorithm operates as if

**Algorithm 1** ModCB (Model Selection for Contextual Bandits)

---

**input:**

- Feature maps $\{\phi^m(\cdot,\cdot)\}_{m\in[M]}$, where $\phi^m(x,a) \in \mathbb{R}^{d_m}$, and time $T \in \mathbb{N}$.
- Subgaussian parameter $\tau > 0$, second moment parameter $\gamma > 0$.
- Failure probability $\delta \in (0,1)$, exploration parameter $\kappa \in (0,1)$, confidence parameters. $C_1, C_2 > 0$.

**definitions:**

- Define $\delta_0 = \delta/10M^2T^2$ and $\mu_t = (K/t)^\kappa \wedge 1$.
- Define $\alpha_{m,t} = C_1 \cdot \left( \frac{\tau^6}{\gamma^4} \cdot \frac{d_m^{1/2}\log^2(2d_m/\delta_0)}{K^\kappa t^{1-\kappa}} + \frac{\tau^{10}}{\gamma^8} \cdot \frac{d_m \log(2/\delta_0)}{t} \right)$.
- Define $T_m^{\min} = C_2 \cdot \left( \frac{\tau^4}{\gamma^2} \cdot d_m \log(2/\delta_0) + \log^{\frac{1}{1-\kappa}}(2/\delta_0) + K \right) + 1$.

**initialization:**

- $\widehat{m} \leftarrow 1$. `// Index of candidate policy class.`
- $\mathsf{Exp4\text{-}IX}_1 \leftarrow \mathsf{Exp4\text{-}IX}(\Pi_1, T, \delta_0)$.
- $S \leftarrow \{\varnothing\}$. `// Times at which uniform exploration takes place.`

**for** $t = 1, \ldots, T$ **do**

    Receive $x_t$.

    **with probability** $1 - \mu_t$

        Feed $x_t$ into $\mathsf{Exp4\text{-}IX}_{\widehat{m}}$ and take $a_t$ to be the predicted action.

        Update $\mathsf{Exp4\text{-}IX}_{\widehat{m}}$ with $(x_t, a_t, \ell_t(a_t))$.

    **otherwise**

        Sample $a_t$ uniformly from $\mathcal{A}$ and let $S \leftarrow S \cup \{t\}$.

    `/* Test whether we should move on to a larger policy class. */`

    $\widehat{\Sigma}_i \leftarrow \frac{1}{K}\sum_{a\in\mathcal{A}}\sum_{s=1}^{t}\phi^i(x_s,a)\phi^i(x_s,a)^\top$ for each $i \geq \widehat{m}$. `// Empirical second moment.`

    $H_i \leftarrow \{(\phi^i(x_s,a_s), \ell(a_s))\}_{s\in S}$ for each $i > \widehat{m}$.

    $\widehat{\mathcal{E}}_{\widehat{m},i} \leftarrow \mathsf{EstimateResidual}(H_i, \widehat{\Sigma}_{\widehat{m}}, \widehat{\Sigma}_i)$ for each $i > \widehat{m}$. `// Gap estimate.`

    **if** there exists $i > \widehat{m}$ such that $\widehat{\mathcal{E}}_{\widehat{m},i} \geq 2\alpha_{i,t}$ and $t \geq T_i^{\min}$ **then**

        Let $\widehat{m}$ be the smallest such $i$. Re-initialize $\mathsf{Exp4\text{-}IX}_{\widehat{m}} \leftarrow \mathsf{Exp4\text{-}IX}(\Pi_{\widehat{m}}, T, \delta_0)$.

---

$\widehat{m} = m^\star$ by running a low-regret contextual bandit algorithm with the policies induced by $\mathcal{F}_{\widehat{m}}$; we call this policy class $\Pi_m := \{x \mapsto \operatorname{argmin}_{a\in\mathcal{A}}\langle\beta, \phi^m(x,a)\rangle \mid \|\beta\|_2 \leq \tau/\gamma\}$.[5] Note that the low-regret algorithm we run for $\Pi_{\widehat{m}}$ cannot based on realizability, since $\mathcal{F}_{\widehat{m}}$ will not contain the true regressor $f^\star$ until we reach $m^\star$. This rules out the usual linear contextual bandit algorithms such as LinUCB. Instead we use a variant of Exp4-IX (Neu, 2015), which is an agnostic contextual bandit algorithm and does not depend on realizability. To deal with infinite classes, unbounded losses, and other technical issues, we require some simple modifications to Exp4-IX; pseudocode and analysis are deferred to Appendix C.3.

### 3.1 Key idea: Estimating prediction error with sublinear sample complexity

Before stating the main result, we elaborate on the statistical test (EstimateResidual) used in Algorithm 1. EstimateResidual estimates an upper bound on the gap between the best-in-class loss for two policy classes $\Pi_i$ and $\Pi_j$, which we define as $\Delta_{i,j} := L_i^\star - L_j^\star$, where $L_i^\star := \min_{\pi\in\Pi_i} L(\pi)$. At each round, Algorithm 1 uses EstimateResidual to estimate the gap $\Delta_{\widehat{m},i}$ for all $i > \widehat{m}$. If the estimated gap is sufficiently large for some $i$, the algorithm sets $\widehat{m}$ to the smallest such $i$ for the next round. This approach is based on the following observation: For all $m \geq m^\star$, $L_m^\star = L_{m^\star}^\star$. Hence, if $\Delta_{\widehat{m},i} > 0$, it must be the case that $\widehat{m} \neq m^\star$, and we should move on to a larger class.

The key challenge is to estimate $\Delta_{\widehat{m},i}$ while ensuring low regret. Naively, we could use uniform exploration and find a policy in $\Pi_i$ that has minimal empirical loss, which gives an estimate of $L_i^\star$. Unfortunately, this requires tuning the exploration parameter in terms of $d_i$ and would compromise the regret if $\widehat{m} = m^\star$. Similar tuning issues arise with other approaches and are discussed further in Appendix B.

**Algorithm 2** EstimateResidual

---

**input:** Examples $\{(x_s, y_s)\}_{s=1}^n$, second moment matrix estimates $\widehat{\Sigma} \in \mathbb{R}^{d \times d}$ and $\widehat{\Sigma}_1 \in \mathbb{R}^{d_1 \times d_1}$. Define $d_2 = d - d_1$ and

$$\widehat{R} = \widehat{D}^\dagger - \widehat{\Sigma}^\dagger, \quad \text{where} \quad \widehat{D} = \begin{pmatrix} \widehat{\Sigma}_1 & 0_{d_1 \times d_2} \\ 0_{d_2 \times d_1} & 0_{d_2 \times d_2} \end{pmatrix}.$$

Return estimator

$$\widehat{\mathcal{E}} = \frac{1}{\binom{n}{2}} \sum_{s < t} \langle \widehat{\Sigma}^{1/2} \widehat{R} x_s y_s, \widehat{\Sigma}_1^{1/2} \widehat{R} x_t y_t \rangle.$$

---

We do not estimate the gaps $\Delta_{i,j}$ directly, but instead estimate an upper bound motivated by the realizability assumption. For each $m$, define

$$\beta_m^\star := \operatorname*{argmin}_{\beta \in \mathbb{R}^{d_m}} \frac{1}{K} \sum_{a \in \mathcal{A}} \mathbb{E}_{x \sim \mathcal{D}} (\langle \beta, \phi^m(x, a) \rangle - \ell(a))^2, \tag{3}$$

and define[6]

$$\mathcal{E}_{i,j} := \frac{1}{K} \sum_{a \in \mathcal{A}} \mathbb{E}_{x \sim \mathcal{D}} \big( \langle \beta_i^\star, \phi^i(x, a) \rangle - \langle \beta_j^\star, \phi^j(x, a) \rangle \big)^2. \tag{4}$$

We call $\mathcal{E}_{i,j}$ the *square loss gap* and we call $\Delta_{i,j}$ the *policy gap*. A key lemma driving these definitions is that the square loss gap upper bounds the policy gap.

**Lemma 1.** *For all $i \in [M]$ and $j \geq m^\star$, $\Delta_{i,j} \leq \sqrt{4K\mathcal{E}_{i,j}}$. Furthermore, if $i, j \geq m^\star$ then $\mathcal{E}_{i,j} = 0$.*

With realizability, the square loss gap behaves similar to the policy gap: it is non-zero whenever the latter is non-zero, and $m^\star$ has zero gap to all $m \geq m^\star$. Therefore, we seek estimators for the square loss gap $\mathcal{E}_{\widehat{m},i}$ for $i > \widehat{m}$. Observe that $\mathcal{E}_{\widehat{m},i}$ depends on the optimal predictors $\beta_{\widehat{m}}^\star, \beta_i^\star$ in the two classes. A natural approach to estimate $\mathcal{E}_{\widehat{m},i}$ is to solve regression problems over both classes to estimate the predictors, then plug them into the expression for $\mathcal{E}$; we call this the *plug-in approach*. As this relies on linear regression, it gives an $O(d_i/n)$ error rate for estimating $\mathcal{E}_{\widehat{m},i}$ from $n$ uniform exploration samples. Unfortunately, since the error scales linearly with the size of the larger class, we must over-explore to ensure low error, and this compromises the regret if the smaller class is optimal.

As a key technical contribution, we design more efficient estimators for the square loss gap $\mathcal{E}_{\widehat{m},i}$. We work in the following slightly more general gap estimation setup: we receive pairs $\{(x_s, y_s)\}_{s=1}^n$ i.i.d. from a distribution $\mathcal{D} \in \Delta(\mathbb{R}^d \times \mathbb{R})$, where $x \sim \mathsf{subG}(\tau^2)$ and $y \sim \mathsf{subG}(\sigma^2)$. We partition the feature space into $x = (x^{(1)}, x^{(2)})$, where $x^{(1)} \in \mathbb{R}^{d_1}$, and define

$$\beta^\star := \operatorname*{argmin}_{\beta \in \mathbb{R}^d} \mathbb{E} \left( \langle \beta, x \rangle - y \right)^2, \qquad \beta_1^\star := \operatorname*{argmin}_{\beta \in \mathbb{R}^{d_1}} \mathbb{E} \left( \langle \beta, x^{(1)} \rangle - y \right)^2.$$

These are, respectively, the optimal linear predictor and the optimal linear predictor restricted to the first $d_1$ dimensions. The square loss gap for the two predictors is defined as $\mathcal{E} := \mathbb{E} \left( \langle \beta^\star, x \rangle - \langle \beta_1^\star, x^{(1)} \rangle \right)^2$. Our problem of estimating $\mathcal{E}_{\widehat{m},i}$ clearly falls into this general setup if we uniformly explore the actions for $n$ rounds, then set $\{x_s\}_{s=1}^n$ to be the features obtained through the feature map $\phi^i$ and $\{y_s\}_{s=1}^n$ to be the observed losses.

The pseudocode for our estimator EstimateResidual is displayed in Algorithm 2. In addition to the $n$ labeled samples, it takes as input two empirical second moment matrices $\widehat{\Sigma}$ and $\widehat{\Sigma}_1$ constructed via an extra set of $m$ i.i.d. unlabeled samples; these serve as estimates for $\Sigma := \mathbb{E}[xx^\top]$ and $\Sigma_1 := \mathbb{E}[x^{(1)}x^{(1)\top}]$. The intuition is that one can write the square loss gap as $\mathcal{E} = \|\Sigma^{1/2} R \mathbb{E}[xy]\|_2^2$ where $R \in \mathbb{R}^{d \times d}$ is a certain function of $\Sigma$ and $\Sigma_1$. EstimateResidual simply replaces the second moment matrices with their empirical counterparts and then uses the labeled examples to estimate the weighted norm of $\mathbb{E}[xy]$ through a U-statistic. The main guarantee for the estimator is as follows.

**Theorem 2.** *Suppose we take $\widehat{\Sigma}$ and $\widehat{\Sigma}_1$ to be the empirical second moment matrices formed from $m$ i.i.d. unlabeled samples. Then when $m \geq C(d + \log(2/\delta))\tau^4/\lambda_{\min}(\Sigma)$,* EstimateResidual, *given $n$ labeled samples, guarantees that with probability at least $1 - \delta$,*

$$\left|\widehat{\mathcal{E}} - \mathcal{E}\right| \leq \frac{1}{2}\mathcal{E} + O\left(\frac{\sigma^2\tau^4}{\lambda_{\min}^2(\Sigma)} \cdot \frac{d^{1/2}\log^2(2d/\delta)}{n} + \frac{\tau^6}{\lambda_{\min}^4(\Sigma)} \cdot \frac{d\log(2/\delta)}{m} \cdot \|\mathbb{E}[xy]\|_2^2\right). \quad (5)$$

To compare with the plug-in approach, we focus on the dependence between $d$ and $n$. When EstimateResidual is applied within ModCB we have plenty of unlabeled data, so the dependence on $m$ is less important. The dominant term in Theorem 2 is $\tilde{O}(\sqrt{d}/n)$, a significant improvement over the $\tilde{O}(d/n)$ rate for the plug-in estimator. In particular, the dependence on the larger ambient dimension is much milder: we can achieve constant error with $n \asymp \sqrt{d}$, or in other words the estimator has *sublinear* sample complexity. This property is crucial for our model selection result. The result generalizes and is inspired by the variance estimation method of Dicker (Dicker, 2014; Kong and Valiant, 2018), which obtains a rate of $O\left(\sqrt{d}/n + 1/\sqrt{n}\right)$ to estimate the optimal square loss $\min_{\beta \in \mathbb{R}^d} \mathbb{E}(\langle\beta, x\rangle - y)^2$ when the second moments are known. By estimating the square loss *gap* instead of the loss itself, we avoid the $1/\sqrt{n}$ term, which is critical for achieving $\tilde{O}(T^{2/3}d_{m^\star}^{1/3})$ regret.

## 3.2 Main result

Equipped with EstimateResidual, we can now sketch the approach behind ModCB in a bit more detail. Recall that the algorithm maintains an index $\widehat{m}$ denoting the current guess for $m^\star$. We run Exp4-IX over the induced policy class $\Pi_m$, mixing in some additional uniform exploration (with probability $\mu_t$ at round $t$). We use all of the data to estimate the second moment matrices of all classes, and we pass only the exploration data into the subroutine EstimateResidual. We check if there exists some $i > \widehat{m}$ such that the estimated gap satisfies $\widehat{\mathcal{E}}_{\widehat{m},i} \geq 2\alpha_{i,t}$ and $t \geq T_i^{\min}$ which—based on the deviation bound in Theorem 2—implies that $\mathcal{E}_{\widehat{m},i} > 0$ and thus $\widehat{m} \neq m^\star$. If this is the case, we advance $\widehat{m}$ to the smallest such $i$, and if not, we continue with our current guess. This leads to the following guarantee.

**Theorem 3.** *When $C_1$ and $C_2$ are sufficiently large absolute constants and $\kappa = 1/3$,* ModCB *guarantees that with probability at least $1 - \delta$,*

$$\mathrm{Reg} \leq \tilde{O}\left(\frac{\tau^4}{\gamma^3} \cdot (Tm^\star)^{2/3}(Kd_{m^\star})^{1/3}\log(2/\delta)\right). \quad (6)$$

*When $\kappa = 1/4$,* ModCB *guarantees that with probability at least $1 - \delta$,*

$$\mathrm{Reg} \leq \tilde{O}\left(\frac{\tau^3}{\gamma^2} \cdot K^{1/4}(Tm^\star)^{3/4}\log(2/\delta) + \frac{\tau^5}{\gamma^4} \cdot \sqrt{K(Tm^\star)d_{m^\star}}\log(2/\delta)\right). \quad (7)$$

A few remarks are in order

- The two stated bounds are incomparable in general. Tracking only dependence on $T$ and $d_{m^\star}$, the first is $\tilde{O}(T^{2/3}d_{m^\star}^{1/3})$ while the latter is $\tilde{O}(T^{3/4} + \sqrt{Td_{m^\star}})$. The former is better when $d_{m^\star} \leq T^{1/4}$. There is a more general Pareto frontier that can be explored by choosing $\kappa \in [1/3, 1/4]$, but no choice for $\kappa$ dominates the others for all values of $d_{m^\star}$.

- Recall that if had we known $d_{m^\star}$ in advance, we have could simply run LinUCB to achieve $\tilde{O}(\sqrt{Td_{m^\star}})$ regret. The bound (7) matches this oracle rate when $d_{m^\star} > \sqrt{T}$, but otherwise our guarantee is slightly worse than the oracle rate. Nevertheless, both bounds are $o(T)$ whenever the oracle rate is $o(T)$ (that is, when $d_{m^\star} = o(T)$), so the algorithm has sublinear regret whenever the optimal model class is *learnable*. It remains open whether there is a model selection algorithm that can match the oracle rate for all values of $d_{m^\star}$ simultaneously.

- We have not optimized dependence on the condition number $\tau/\gamma$ or the logarithmic factors.

- If the individual distribution parameters $\{\tau_m\}_{m \in [M]}$ and $\{\gamma_m\}_{m \in [M]}$ are known, the algorithm can be modified slightly so that regret scales in terms of $\tau_{m^\star}$ and $\gamma_{m^\star}^{-1}$. However the current model, in which we assume access only to uniform upper and lower bounds on these parameters, is more realistic.

Additionally, Appendix A contains a validation experiment, where we demonstrate that ModCB compares favorably to LinUCB in simulations.

**Improving the dependence on $m^\star$.** Theorem 3 obtains the desired model selection guarantee for linear classes, but the bound includes a polynomial dependence on the optimal index $m^\star$. This contrasts the logarithmic dependence on $m^\star$ that can be obtained through structural risk minimization in statistical learning (Vapnik, 1992). However, this $\mathrm{poly}(m^\star)$ dependence can be replaced by a single $\log(T)$ factor with a simple preprocessing step: Given feature maps $\{\phi^m(\cdot,\cdot)\}_{m\in[M]}$ we construct a new collection of maps $\{\bar\phi^m(\cdot,\cdot)\}_{m\in[\bar M]}$, where $\bar M \le \log T$, as follows. First, for $i = 1,\ldots,\log T$, take $\bar\phi^i$ to be the largest feature map $\phi^m$ for which $d_m \le e^i$. Second, remove any duplicates. This preprocessing reduces the number of feature maps to at most $\log(T)$ while ensuring that a map of dimension $O(d_{m^\star})$ that contains $\phi^{m^\star}$ is always available. Specifically, the preprocessing step yields the following improved regret bounds.

**Theorem 4.** *ModCB with preprocessing guarantees that with probability at least $1 - \delta$,*

$$\mathrm{Reg} \le \begin{cases} \tilde O\!\left(\frac{\tau^4}{\gamma^3}\cdot T^{2/3}(Kd_{m^\star})^{1/3}\log(2/\delta)\right), & \kappa = 1/3. \\[2mm] \tilde O\!\left(\frac{\tau^3}{\gamma^2}\cdot K^{1/4}T^{3/4}\log(2/\delta) + \frac{\tau^5}{\gamma^4}\cdot\sqrt{KTd_{m^\star}}\log(2/\delta)\right), & \kappa = 1/4. \end{cases}$$

**Non-nested feature maps.** As a final variant, we note that the algorithm easily extends to the case where feature maps are not nested. Suppose we have non-nested feature maps $\{\phi^m\}_{m\in[M]}$, where $d_1 \le d_2 \le \ldots \le d_M$; note that the inequality is no longer strict. In this case, we can obtain a nested collection by concatenating $\phi^1,\ldots,\phi^{m-1}$ to the map $\phi^m$ for each $m$. This process increases the dimension of the optimal map from $d_{m^\star}$ to at most $d_{m^\star}m^\star$, so we have the following corollary.

**Corollary 5.** *For non-nested feature maps, ModCB with preprocessing guarantees that with probability at least $1 - \delta$,*

$$\mathrm{Reg} \le \begin{cases} \tilde O\!\left(\frac{\tau^4}{\gamma^3}\cdot T^{2/3}(Kd_{m^\star}m^\star)^{1/3}\log(2/\delta)\right), & \kappa = 1/3. \\[2mm] \tilde O\!\left(\frac{\tau^3}{\gamma^2}\cdot K^{1/4}T^{3/4}\log(2/\delta) + \frac{\tau^5}{\gamma^4}\cdot\sqrt{KTd_{m^\star}m^\star}\log(2/\delta)\right), & \kappa = 1/4. \end{cases}$$

### 3.3 Proof sketch

We now sketch the proof of the main theorem, with the full proof deferred to Appendix D. We focus on the case where there are just two classes, so $M = 2$. We only track dependence on $T$ and $d_m$, as this preserves the relevant details but simplifies the argument. The analysis has two cases depending on whether $f^\star$ belongs to $\mathcal{F}_1$ or $\mathcal{F}_2$.

First, if $f^\star \in \mathcal{F}_1$ then by Lemma 1 we have that $\mathcal{E}_{1,2} = 0$. Further, via Theorem 2, we can guarantee that we never advance to $\widehat m = 2$ with high probability. The result then follows from the regret guarantee for Exp4-IX using policy class $\Pi_1$, and by accounting for uniform exploration.

The more challenging case is when $f^\star \in \mathcal{F}_2$. Let $\widehat T$ denote the first round where $\widehat m = 2$, or $T$ if the algorithm never advances. We can bound regret as

$$\mathrm{Reg} \le O\!\left(T^{1-\kappa}\right) + \tilde O\!\left(\sqrt{\widehat T d_1}\right) + \widehat T\Delta_{1,2} + \tilde O\!\left(\sqrt{(T - \widehat T)d_2}\right).$$

The four terms correspond to: (1) uniform exploration with probability $\mu_t \asymp t^{-\kappa}$ in round $t$, (2) the Exp4-IX regret bound for class $\Pi_1$ until time $\widehat T$, (3) the policy gap between the best policy in $\Pi_1$ and the optimal policy $\pi^\star \in \Pi_2$, and (4) the Exp4-IX bound over class $\Pi_2$ until time $T$. The two regret bounds (the second and fourth terms) clearly contribute $O(\sqrt{Td_2})$ to the overall regret, and the first term is controlled by our choice of $\kappa \in \{1/4, 1/3\}$. We are left to bound the third term. For this term, observe that in round $\widehat T - 1$, since the algorithm did not advance, we must have $\widehat{\mathcal{E}}_{1,2} \le 2\alpha_{2,\widehat T - 1}$. Appealing to Theorem 2, this implies that $\mathcal{E}_{1,2} \le O(\widehat{\mathcal{E}}_{1,2} + \alpha_{2,\widehat T - 1}) \le O(\alpha_{2,\widehat T - 1})$. Plugging in the definition of $\alpha_{2,t}$ and applying Lemma 1, this gives

$$\widehat T\Delta_{1,2} \le O\!\left(\widehat T\sqrt{\mathcal{E}_{1,2}}\right) \le O\!\left(\widehat T\sqrt{\alpha_{2,\widehat T - 1}}\right) = O\!\left(T^{\frac{1+\kappa}{2}}d_2^{1/4}\right). \tag{8}$$

This establishes the result. In particular, with $\kappa = 1/3$ we obtain $\tilde{O}(T^{2/3} + \sqrt{Td_2} + T^{2/3}d_2^{1/4}) \leq \tilde{O}(T^{2/3}d_2^{1/3})$ regret, and with $\kappa = 1/4$ we obtain $\tilde{O}(T^{3/4} + \sqrt{Td_2} + T^{5/8}d_2^{1/4}) = \tilde{O}(T^{3/4} + \sqrt{Td_2})$.[7]

The sublinear estimation rate for EstimateResidual (Theorem 2) plays a critical role in this argument. With the $\tilde{O}(d/n)$ rate for the naïve plug-in estimator, we could at best set $\alpha_{t,2} = d_2/t^{1-\kappa}$, but this would degrade the dimension dependence in (8) from $d_2^{1/4}$ to $\sqrt{d_2}$. Unfortunately, this results in $T^{1-\kappa} + \sqrt{d_2}T^{\frac{1+\kappa}{2}}$ regret, which is not a meaningful model selection result, since there is no choice for $\kappa \in (0,1)$ for which the exponents on $d_2$ and $T$ sum to one for both terms.

## 4 Discussion

This paper initiates the study of model selection tradeoffs in contextual bandits. We provide the first model selection algorithm for the linear contextual bandit setting, which we show achieves $\tilde{O}\left(T^{2/3}d_{m^\star}^{1/3}\right)$ when the optimal model is a $d_{m^\star}$-dimensional linear function. This is the first contextual bandit algorithm that adapts the complexity of the optimal policy class with no prior knowledge, and raises a number of intriguing questions:

1. Is it possible to achieve $\tilde{O}\left(\sqrt{Td_{m^\star}}\right)$ regret in our setup? This would show that the price for model selection is negligible.

2. Is it possible to generalize our results beyond linear classes? Specifically, given regressor classes $\mathcal{F}_1 \subset \mathcal{F}_2 \subset \ldots \subset \mathcal{F}_M$ and assuming the optimal model $f^\star$ belongs to $\mathcal{F}_{m^\star}$ for some unknown $m^\star$, is there a contextual bandit algorithm that achieve $\tilde{O}\left(T^\alpha \cdot \text{comp}^{1-\alpha}(\mathcal{F}_{m^\star})\right)$ regret for some $\alpha \in [1/2, 1)$? More ambitiously, can we achieve $\tilde{O}\left(\sqrt{T \cdot \text{comp}(\mathcal{F}_{m^\star})}\right)$?

3. We have conducted a validation experiment with ModCB (see Appendix A), demonstrating that the algorithm performs favorably in simulations. While this synthetic experiment is encouraging, ModCB may not be immediately useful for practical deployments for several reasons, including its reliance on linear realizability and its computational complexity. Are there more robust algorithmic principles for theoretically-sound and practically-effective model selection in contextual bandits?

4. For what classes $\mathcal{F}$ can we estimate the loss at a sublinear rate, and is this necessary for contextual bandit model selection? Any sublinear guarantee will lead to non-trivial model selection guarantees through a strategy similar to ModCB. Interestingly, it is already known that for certain (e.g., sparse linear) classes, sublinear loss estimation is not possible (Verzelen and Gassiat, 2018). On the other hand, positive results *are* available for certain nonparametric classes (Brown et al., 2007; Wang et al., 2008).

Model selection is instrumental to the success of ML deployments, yet few positive results exist for partial feedback settings. We believe these questions are technically challenging and practically important, and we are hopeful that positive results of the type we provide here will extend to interactive learning settings beyond contextual bandits.

#### Acknowledgements

We thank Ruihao Zhu for working with us at the early stages of this project, and for many helpful discussions. AK is supported by NSF IIS-1763618. HL is supported by NSF IIS-1755781.

## Footnotes

[1] We suppress dependence on $K$ and logarithmic dependence on $T$ for this discussion.

[2]This is without loss of generality in a certain quantitative sense, since we can concatenate features without significantly increasing the complexity of $\mathcal{F}_m$. See Corollary 5.

[3]Regret scaling as $\tilde{O}(\sqrt{dT})$ is optimal for the finite action setting we work in. Results for the infinite action case, where regret scales as $\tilde{\Theta}(d\sqrt{T})$, are incomparable to ours.

[4]It appears that exploration is still required for linear contextual bandits under our average eigenvalue assumption. Consider the case $d = 2$ and $\beta^\star = (1/2, 1)$. Suppose there are four actions, and that at the first round, $\phi(x, \cdot) = \{e_1, -e_1, e_2, -e_2\}$. We can ensure that with probability $1/2$, the first action played will be one of the first two. At this point a greedy strategy will always choose $e_1$, but the average context distribution has minimum eigenvalue 1.

[5]The norm constraint $\tau/\gamma$ guarantees that $\Pi_m$ contains parameter vectors arising from a certain square loss minimization problem under our assumption that $\|\beta^\star\|_2 \leq 1$; see Proposition 20.

[6]In Appendix D we show that $\beta_m^\star$ and consequently $\mathcal{E}_{i,j}$ are always uniquely defined.

[7]Note that if $d_2 \leq \sqrt{T}$ then $T^{5/8}d_2^{1/4} \leq T^{3/4}$, but if $d_2 \geq \sqrt{T}$ then $T^{5/8}d_2^{1/4} \leq \sqrt{Td_2}$.

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
