[Supplementary Material · supplementary.pdf]

# A  A validation experiment

As an empirical validation, we conducted a synthetic experiment with ModCB. Our implementation is built on top of an open source package for contextual bandit experimentation, which has been used in several prior works (Krishnamurthy et al., 2016; Foster et al., 2018; Krishnamurthy et al., 2018).[8] For computational efficiency, our implementation of ModCB uses ILOVETOCONBANDITS (Agarwal et al. (2014), henceforth "ILTCB") as the base learner instead of Exp4-IX, which is also sufficient for our theoretical guarantees.

We consider a simple contextual bandit environment with $K = 2$ actions, and where the feature vectors have ambient dimension $d = 1000$. We design the reward distribution such that a predictor using the first $d_{m^\star} = 10$ coordinates is realizable. We consider three algorithms: LinUCB operating directly on the ambient dimension, ModCB with ILTCB as the base learner, and ILTCB with knowledge of $d_{m^\star}$, which we refer to as Oracle. We tune a single hyperparameter for each algorithm (the confidence pre-multiplier for LinUCB; the uniform exploration parameter for ModCB and Oracle) and visualize the cumulative regret for the

Figure 1: A validation experiment

best performing hyperparameter as a function of the number of rounds $T$. Figure 1 displays the regret curves averaged over 20 replicates with error bands corresponding to two standard errors.

ModCB consistently outperforms LinUCB in the experiment, which is unsurprising since the ambient dimension is much larger than the target dimension $d_{m^\star}$. ModCB has a less favorable dependence on $T$ in comparison to LinUCB, but this does not seem to compromise its performance in this experiment. Perhaps more surprising is that ModCB outperforms Oracle on average. A deeper inspection reveals that while ModCB typically advances to $d_m > d_{m^\star}$, it sometimes stays below, where it can learn a near-optimal policy faster than Oracle.

# B  Omitted details for Section 2

In this section we provide more detail as to why various natural approaches do not provide a satisfactory resolution to the model selection problem for contextual bandits. We consider a more general setup here, where the learner is given a set of policy classes $\Pi_1, \ldots, \Pi_M$, each of which contains a set of arbitrary mappings from the context space $\mathcal{X}$ to $\mathcal{A}$. The (expected) regret to class $m$ is

$$\mathrm{Reg}(\Pi_m) := \sum_{t=1}^{T} \mathbb{E}\left[\ell_t(a_t)\right] - \min_{\pi \in \Pi_m} \sum_{t=1}^{T} \mathbb{E}\left[\ell_t(\pi(x_t))\right].$$

Let $m^\star := \mathrm{argmin}_m \min_{\pi \in \Pi_m} \sum_{t=1}^{T} \mathbb{E}\left[\ell_t(\pi(x_t))\right]$ be the index of the class containing the optimal policy. The goal is to achieve $\mathrm{Reg}(\Pi_{m^\star}) = O\left(T^\alpha \cdot \mathrm{comp}^{1-\alpha}(\Pi_{m^\star})\right)$ for some $\alpha \in [1/2, 1)$ without knowing $m^\star$ ahead of time (ignoring the dependence on $K$). Our realizable linear setting is clearly a special case of this setup. It is well-known that the model selection guarantee we desire can be achieved in the full-information online learning setting, and below we discuss the difficulties of extending these approaches to the bandit setting, even when $M = 2$. For simplicity we consider finite classes, so the complexity of $\Pi_m$ is measured by $\log |\Pi_m|$.

## B.1  Running Hedge with all policy classes

The classical contextual bandit algorithm Exp4 (Auer et al., 2002) is based on the full-information algorithm Hedge (Freund and Schapire, 1997). Fix a policy class $\Pi$. At each time $t$, Exp4 computes a distribution $p_t$ over the policies in $\Pi$ using the exponential weight update rule:

$$p_t(\pi) \propto p_0(\pi) \exp\left(\eta \sum_{\tau < t} \hat{\ell}_\tau(\pi(x_\tau))\right),$$

where $\eta$ is a learning rate parameter, $p_0$ is a prespecified prior distribution over the policies, and $\hat{\ell}_\tau$ is the importance-weighted loss estimator, defined as $\hat{\ell}_\tau(a) := \frac{\ell_\tau(a)\mathbb{I}\{a_\tau = a\}}{\sum_{\pi \in \Pi : \pi(x_\tau) = a} p_\tau(\pi)}$ for all $a$. Exp4 ensures for any $\pi^\star \in \Pi$,

$$\sum_{t=1}^{T} \mathbb{E}\left[\ell_t(a_t)\right] - \sum_{t=1}^{T} \mathbb{E}\left[\ell_t(\pi^\star(x_t))\right] \leq \frac{\log\left(\frac{1}{p_0(\pi^\star)}\right)}{\eta} + \eta \mathbb{E}\left[\sum_{t=1}^{T} \sum_{\pi \in \Pi} p_t(\pi)\hat{\ell}_t(\pi(x_t))^2\right] \qquad (9)$$

$$\leq \frac{\log\left(\frac{1}{p_0(\pi^\star)}\right)}{\eta} + \eta T K.$$

Now, consider running this algorithm with $\Pi := \bigcup_{m=1}^{M} \Pi_m$. With a uniform prior and the optimal tuning of $\eta$, this leads the following regret bound for all $m$, which is clearly undesirable:

$$\text{Reg}(\Pi_m) = O\left(\sqrt{TK \log\left(\sum_{m=1}^{M} |\Pi_m|\right)}\right).$$

On the other hand, if we set $p_0(\pi) := 1/(M|\Pi_m|)$, where $m$ is the least index such that $\pi \in \Pi_m$, then we have that for each $m$,

$$\text{Reg}(\Pi_m) \leq \frac{\log\left(M|\Pi_m|\right)}{\eta} + \eta T K.$$

With oracle tuning for $\eta$ this would give $\text{Reg}(\Pi_{m^\star}) = \text{Reg} \leq O(\sqrt{TK \log|\Pi_{m^\star}|})$, as desired. The issue, of course, is that tuning requires knowing $m^\star$ ahead of time. One can instead simply set $\eta = 1/\sqrt{TK}$ and obtain for all $m$,

$$\text{Reg}(\Pi_m) = O\left(\sqrt{TK} \log\left(M|\Pi_m|\right)\right),$$

which is not a satisfactory model selection guarantee. In particular this bound is vacuous when $\log|\Pi_{m^\star}| = \Omega(\sqrt{T})$, even though we could have achieved sublinear regret here if $m^\star$ were known.

A natural next step would be to apply an individual learning rate $\eta_m$ for each class separately, with the hope of achieving for all $m$,

$$\text{Reg}(\Pi_m) = \frac{\log\left(M|\Pi_m|\right)}{\eta_m} + \eta_m T K,$$

which will then solve the problem by setting $\eta_m = \sqrt{\log(|\Pi_m|)/(TK)}$. However, we are not aware of any existing approaches that achieve this guarantee. The closest guarantee (achieved by variants of Hedge (Gaillard et al., 2014; Koolen and Van Erven, 2015)) is that for all $m$ and $\pi_m \in \Pi_m$,

$$\sum_{t=1}^{T} \mathbb{E}\left[\ell_t(a_t)\right] - \sum_{t=1}^{T} \mathbb{E}\left[\ell_t(\pi_m(x_t))\right] \leq \frac{\log\left(M|\Pi_m|\right)}{\eta_m} + \eta_m \mathbb{E}\left[\sum_{t=1}^{T}(\ell_t(a_t) - \hat{\ell}_t(\pi_m(x_t)))^2\right]$$

$$\leq \frac{\log\left(M|\Pi_m|\right)}{\eta_m} + 2\eta_m T + 2\eta_m \mathbb{E}\left[\sum_{t=1}^{T} \hat{\ell}_t(\pi_m(x_t))^2\right].$$

Unfortunately, this does not enjoy the useful local norm term as in (9), and in particular the last term involving the second moment of the loss estimator can be unbounded. Common fixes such as forcing a small amount of uniform exploration all lead to further tuning issues.

## B.2  Aggregating via Corral

Next, we briefly describe issues with using Corral (Agarwal et al., 2017b) to aggregate multiple instances of Exp4. In short, using Theorem 4 of (Agarwal et al., 2017b), one can verify that the algorithm ensures for all $m$,

$$\text{Reg}(\Pi_m) = \tilde{O}\left(\frac{M}{\eta} + T\eta - \frac{\rho_m}{\eta} + \sqrt{TK \log(|\Pi_m|)\rho_m}\right),$$

for a fixed parameter $\eta$ and a certain data-dependent quantity $\rho_m$. Using the AM-GM inequality to upper bound the last term by $\eta TK \log(|\Pi_m|) + \rho_m/\eta$, and canceling with the third term, we have $\text{Reg}(\Pi_m) = \tilde{O}\left(\frac{M}{\eta} + \eta TK \log(|\Pi_m|)\right)$. At this point, one can see that the tuning issues similar to those discussed in the previous section appear, and again there is no obvious fix.

## B.3 Adaptive $\epsilon$-greedy

Here we present a natural adaptation of an $\epsilon$-greedy algorithm for model selection with two classes $|\Pi_1| \ll |\Pi_2|$. The algorithm does not achieve a satisfactory model selection guarantee, but we include the analysis because it demonstrates some of the difficulties with parameter tuning, even for algorithms based on naive exploration where the best rate one could hope to achieve is $O(T^{2/3}\mathrm{comp}(\mathcal{F}_{m^\star})^{1/3})$.

---

**Algorithm 3** An adaptive explore-first algorithm.

---

Inputs: $T$, policy classes $\Pi_1, \Pi_2$ with $|\Pi_1| \le |\Pi_2|$.

Set $t_1 := \lceil T^{2/3}(K \log |\Pi_1|)^{1/3} \rceil$, $t_2 := \lceil T^{2/3}(K \log |\Pi_2|)^{1/3} \rceil$.

Set $\Delta := \Omega\left( \left( \frac{K}{T \log(T|\Pi_1|)} \right)^{1/3} \sqrt{\log(T|\Pi_2|)} \right)$.

**for** $t = 1, \dots, t_1$ **do**

    Observe $x_t$ choose $a_t \sim \mathrm{Unif}(\mathcal{A})$, observe $\ell_t(a_t)$.

Define $\hat{L}_1 : \pi \mapsto \sum_{t=1}^{t_1} \ell_t(a_t)\mathbf{1}\{\pi(x_t) = a_t\}$. Set

$$\hat{\pi}_1 := \operatorname*{argmin}_{\pi \in \Pi_1} \hat{L}_1(\pi), \qquad \hat{\pi}_2 := \operatorname*{argmin}_{\pi \in \Pi_2} \hat{L}_1(\pi)$$

**if** $\hat{L}_1(\hat{\pi}_1) \le \hat{L}_1(\hat{\pi}_2) + \Delta$ **then**

    Use $\hat{\pi}_1$ to select actions for the rest of time.

**else**

    Explore uniformly for a total of $t_2$ rounds, let $\hat{\pi}_2^{(2)} := \operatorname{argmin}_{\pi \in \Pi_2} \sum_{t=1}^{t_2} \ell_t(a_t)\mathbf{1}\{\pi(x_t) = a_t\}$,

    and use $\hat{\pi}_2^{(2)}$ to select actions for the rest of time.

---

Pseudocode is displayed in Algorithm 3. The algorithm operates in the stochastic setting, where we have $(x_t, \ell_t) \sim \mathcal{D}$ on each round for some distribution $\mathcal{D}$. Define $L(\pi) := \mathbb{E}_{(x,\ell) \sim \mathcal{D}}[\ell(\pi(x))]$, and let $\pi_i^\star := \operatorname{argmin}_{\pi \in \Pi_i} L(\pi)$. We assume that losses are bounded in $[0, 1]$.

The algorithm consists of an exploration phase, a statistical test, a second exploration phase depending on the outcome of the test, and then an exploitation phase. The intuition is that we first explore for $t_1$ rounds, where $t_1$ is the optimal hyperparameter choice for the smaller class $\Pi_1$ (smaller classes require less exploration). Then, we perform a statistical test to determine if $\Pi_1$ can achieve loss that is competitive with $\Pi_2$. If this is the case, we simply exploit $\Pi_1$ for the remaining rounds. Otherwise, we continue exploring for a total of $t_2$ rounds, where $t_2$ is the optimal hyperparameter for $\Pi_2$. We finish by exploiting with the empirical risk minimizer for $\Pi_2$.

**Proposition 6.** *In the stochastic setting, Algorithm 3 achieves the following guarantees simultaneously with probability at least $1 - 1/T$:*

$$\mathrm{Reg}(\Pi_1) \le \tilde{O}\left( T^{2/3}(K \log |\Pi_1|)^{1/3} \right), \quad and \quad \mathrm{Reg}(\Pi_2) \le \tilde{O}\left( T^{2/3} K^{1/3} \frac{\log^{1/2}|\Pi_2|}{\log^{1/3}|\Pi_1|} \right).$$

Note that this is *not* a satisfactory model selection guarantee, since the exponents on $T$ and the policy complexity for $\Pi_2$ do not sum to 1 as we would like. Conceptually, if the algorithm exploits with a fixed policy, it must first determine whether $\Pi_2$ offers much lower loss than $\Pi_1$ so that it can decide which class to use for exploitation. To make this determination, it must estimate the optimal loss for both classes. Unfortunately, with too little exploration data, we will significantly underestimate the loss for $\Pi_2$, and with too much data we will compromise the regret bound for $\Pi_1$. We are not aware of an approach to balance these competing objectives with this style of algorithm.

**Proof of Proposition 6.** Define $\hat{L}_2 : \pi \mapsto \sum_{t=1}^{t_2} \ell_t(a_t)\mathbf{1}\{\pi(x_t) = a_t\}$. We will only use $\hat{L}_2$ in the event that we continue to explore after the test. Via Bernstein's inequality (and using that the deviation is at most 1), the following inequalities all hold with probability at least $1 - \delta$:

$$\forall j \in [2], \forall \pi \in \Pi_i : \left| L(\pi) - \hat{L}_j(\pi) \right| \le c\sqrt{\frac{K \log(|\Pi_i|/\delta)}{t_j}},$$

$$\forall i \in [2] : \left| L(\pi_i^\star) - \hat{L}_1(\pi_i^\star) \right| \le c\sqrt{\frac{K \log(1/\delta)}{t_1}}.$$

Note that the second inequality does not use uniform convergence, and it applies only to each $\pi_i^\star$. Appealing to the standard explore-first analysis, we know that the regret bound for $\Pi_1$ is achieved if we exploit with $\hat{\pi}_1$, and similarly for $\Pi_2$ if we exploit with $\hat{\pi}_2^{(2)}$. We are left to verify the other two cases. Let us consider $\text{Reg}(\Pi_2)$ when we only explore for $t_1$ rounds. We have

$$\text{Reg}(\Pi_2) = t_1 + (T - t_1)\left(L(\hat{\pi}_1) - L(\pi_2^\star)\right)$$

$$\leq t_1 + T\left(c\sqrt{\frac{K\log(|\Pi_1|/\delta)}{t_1}} + \Delta + c\sqrt{\frac{K\log(1/\delta)}{t_1}}\right)$$

$$\leq t_1 + 2c\sqrt{\frac{KT\log(|\Pi_1|/\delta)}{t_1}} + T\Delta.$$

Here we use the two concentration inequalities above, the fact that $\hat{L}_1(\pi_2^\star) \geq \hat{L}_1(\hat{\pi}_2)$, and the fact that the test succeeded. Note that we did not require uniform convergence over $\Pi_2$ for this argument. Now, let us consider the regret for $\Pi_1$ when we explore for $t_2$ rounds.

$\text{Reg}(\Pi_1)$

$$= t_2 + (T - t_2)\left(L(\hat{\pi}_2^{(2)}) - L(\pi_1^\star)\right)$$

$$\leq t_2 + (T - t_2)\left(c\sqrt{\frac{K\log(|\Pi_2|/\delta)}{t_2}} + L(\pi_2^\star) - \hat{L}_1(\hat{\pi}_1) + c\sqrt{\frac{K\log(1/\delta)}{t_1}}\right)$$

$$\leq t_2 + (T - t_2)\left(c\sqrt{\frac{K\log(|\Pi_2|/\delta)}{t_2}} + c\sqrt{\frac{K\log(|\Pi_2|/\delta)}{t_1}} + \hat{L}_1(\hat{\pi}_2) - \hat{L}_1(\hat{\pi}_1) + c\sqrt{\frac{K\log(1/\delta)}{t_1}}\right)$$

$$\leq 4c(T - t_2)\sqrt{\frac{K\log(|\Pi_2|/\delta)}{t_1}} - (T - t_2)\Delta,$$

where we have used that $t_2$ is lower order than the deviation bound term, since $t_1$ is much smaller. To obtain the claimed regret bound for $\Pi_1$, we must choose $\Delta$ as

$$\Delta \geq 4c\sqrt{\frac{K\log(|\Pi_2|/\delta)}{t_1}} - \Omega\left((K\log(|\Pi_1|/\delta)/T)^{1/3}\right) = \Omega\left(\left(\frac{K}{T\log(|\Pi_1|/\delta)}\right)^{1/3}\sqrt{\log(|\Pi_2|/\delta)}\right),$$

with $\delta = 1/T$. $\qquad\qquad\square$

## C  Preliminaries for main results

This section consists of self-contained technical results used to prove the main theorem.

### C.1  Properties of subgaussian and subexponential random variables

Here we state some standard facts about subgaussian and subexponential random variables that will be used throughout the analysis. The reader may consult e.g., Vershynin (2012), for proofs.

Note that if $z \sim \mathsf{subG}_d(\tau^2)$ then we clearly have $\langle z, \theta \rangle \sim \mathsf{subG}(\tau^2\|\theta\|_2^2)$ and likewise if $z \sim \mathsf{subE}_d(\lambda)$ then $\langle z, \theta \rangle \sim \mathsf{subE}(\lambda\|\theta\|_2)$. Furthermore, if $z \sim \mathsf{subG}_d(\sigma^2)$, then $\mathbb{E}[zz^\top] \preceq \sigma^2 \cdot I_d$.

**Proposition 7.** *There exist universal constants $c_1, c_2, c_3 > 0$ such that for any random variable $z \sim \mathsf{subG}(\sigma^2)$, the following hold:*

- $\Pr(|z| > t) \leq 2e^{-\frac{t^2}{c_1\sigma^2}} \quad \forall t \geq 0.$

- $\mathbb{E}[\exp(c_2|z|^2/\sigma^2)] \leq 2.$

- *If $\mathbb{E}[z] = 0$, $\mathbb{E}[\exp(sz)] \leq \exp\left(c_3s^2\sigma^2\right) \quad \forall s \in \mathbb{R}^d.$*

*Moreover, there is some universal constant $c_4$ such that if any of the above properties hold, then $z \sim \mathsf{subG}(c_4\sigma^2)$*

**Proposition 8.** *There exist universal constants $c_1, c_2, c_3 > 0$ such that for any random variable $z \sim \mathsf{subE}(\lambda)$, the following hold:*

- $\Pr(|z| > t) \le 2e^{-\frac{t}{c_1 \lambda}} \quad \forall t \ge 0.$

- *If* $\mathbb{E}[z] = 0$, $\mathbb{E}[\exp(sz)] \le \exp\left(c_2 s^2 \lambda^2\right) \quad \forall s : |s| \le \frac{1}{c_3 \lambda}.$

*Moreover, there is some universal constant $c_4$ such that if any of the above properties hold, then $z \sim \mathsf{subE}(c_4 \lambda)$*

**Proposition 9.** *There is a universal constant $c > 0$ such that the following hold:*

- *If $z \sim \mathsf{subG}(\sigma^2)$, $z - \mathbb{E}[z] \sim \mathsf{subG}(4\sigma^2)$, and if $z \sim \mathsf{subE}(\lambda)$, $z - \mathbb{E}[z] \sim \mathsf{subE}(2\lambda)$.*

- *If $z \sim \mathsf{subG}(\sigma^2)$, then $z^2 - \mathbb{E}[z^2] \sim \mathsf{subE}(c \cdot \sigma^2)$.*

- *If $z_1 \sim \mathsf{subG}(\sigma_1^2)$ and $z_2 \sim \mathsf{subG}(\sigma_2^2)$, then $z_1 z_2 - \mathbb{E}[z_1 z_2] \sim \mathsf{subE}(c \cdot \sigma_1 \sigma_2)$.*

**Proposition 10** (Bernstein's inequality for subexponential random variables)**.** *Let $z_1, \ldots, z_n$ be independent mean-zero random variables with $z_i \sim \mathsf{subE}(\lambda)$ for all $i$, and let $Z = \sum_{i=1}^{n} z_i$. Then for some universal constant $c > 0$,*

$$\Pr(|Z| \ge t) \le 2\exp\left(-c\left(\frac{t^2}{\lambda^2 n} \wedge \frac{t}{\lambda}\right)\right).$$

*In particular, with probability at least $1 - \delta$, $|Z| \le O\left(\sqrt{n\lambda^2 \log(2/\delta)} + \lambda \log(2/\delta)\right).$*

**Lemma 11.** *Let $z_1, \ldots z_n$ be i.i.d. draws of a mean-zero random variable $z \sim \mathsf{subE}_d(\lambda)$, and let $Z_i = \sum_{t=1}^{i} z_t$. Then with probability at least $1 - \delta$,*

$$\max_{i \in [n]} \|Z_i\|_2 \le O\left(\lambda \sqrt{dn \log(2d/\delta)} + \lambda d^{1/2} \log(2d/\delta)\right).$$

**Proof of Lemma 11.** We first claim that for any $s > 0$, the sequence $X_i := e^{s\|Z_i\|_2}$ is a non-negative submartingale. Indeed, using Jensen's inequality, we have

$$\mathbb{E}[X_i \mid z_1, \ldots, z_{i-1}] = \mathbb{E}\left[e^{s\|\Sigma_{t=1}^{i} z_t\|_2} \mid z_1, \ldots, z_{i-1}\right] \ge \mathbb{E}\left[e^{s\|\Sigma_{t=1}^{i-1} z_t\|_2} \mid z_1, \ldots, z_{i-1}\right] = X_{i-1}.$$

Thus, applying the Chernoff method along with Doob's maximal inequality, we have

$$\Pr\left(\max_{i \in [n]} \|Z_i\|_2 > t\right) \le \mathbb{E}\left[\exp\left(s\left\|\sum_{t=1}^{n} z_t\right\|_2 - st\right)\right].$$

We apply the bound

$$\mathbb{E}\left[\exp\left(s\left\|\sum_{t=1}^{n} z_t\right\|_2\right)\right] \le \mathbb{E}\left[\exp\left(s\sqrt{d}\left\|\sum_{t=1}^{n} z_t\right\|_\infty\right)\right] \le \sum_{k=1}^{d} \mathbb{E}\left[\exp\left(s\sqrt{d}\left|\left\langle\sum_{t=1}^{n} z_t, e_k\right\rangle\right|\right)\right],$$

where $e_k$ is the $k$th standard basis vector. Since $z_t \sim \mathsf{subE}_d(\lambda)$ and $\{z_t\}$ are independent, the latter quantity is bounded by

$$2d \cdot \exp\left(Cs^2 d\lambda^2 n\right),$$

so long as $s \le 1/c\lambda\sqrt{d}$, for absolute constants $C, c > 0$. We set $s \propto \frac{t}{d\lambda^2 n} \wedge \frac{1}{\lambda\sqrt{d}}$ to conclude

$$\Pr\left(\max_{i \in [n]} \|Z_i\|_2 > t\right) \le 2d \cdot e^{-C\left(\frac{t^2}{d\lambda^2 n} \vee \frac{t}{\lambda\sqrt{d}}\right)}.$$

Or in other words, with probability at least $1 - \delta$,

$$\max_{i \in [n]} \|Z_i\|_2 \le O\left(\lambda\sqrt{dn \log(2d/\delta)} + \lambda d^{1/2} \log(2d/\delta)\right). \qquad \square$$

## C.2 Second moment matrix estimation

In this section we give some standard results for the rate at which the empirical correlation matrix approaches the population correlation matrix for subgaussian random variables. Let $x \sim \mathsf{subG}_d(\tau^2)$ be a subgaussian random variable and let $\Sigma = \mathbb{E}[xx^\top]$. Let $x_1, \ldots, x_n$ be i.i.d. draws from $x$, and let $\widehat{\Sigma} = \frac{1}{n} \sum_{t=1}^n x_t x_t^\top$.

**Proposition 12.** *Suppose $x \sim \mathsf{subG}_d(\tau^2)$ and $\Sigma = I$. Then with probability at least $1 - \delta$,*

$$1 - \varepsilon \le \lambda_{\min}^{1/2}(\widehat{\Sigma}) \le \lambda_{\max}^{1/2}(\widehat{\Sigma}) \le 1 + \varepsilon, \tag{10}$$

*where $\varepsilon \le 50\tau^2 \sqrt{\frac{d + \log(2/\delta)}{n}}$.*

**Proof of Proposition 12.** Tracking constants carefully, equation (5.23) in Vershynin (2012) establishes that with probability at least $1 - 2e^{-\frac{ct^2}{\tau^4}}$,

$$\left\| \widehat{\Sigma} - I \right\|_2 \le \varepsilon \vee \varepsilon^2,$$

where $\varepsilon = C\tau^2 \sqrt{\frac{d}{n}} + \frac{t}{\sqrt{n}}$, and $c > 10^{-3}$ and $C \le 25$ are absolute constants. The result follows by Vershynin (2012), Lemma 5.36. $\qquad\square$

**Proposition 13.** *Let $\Sigma$ be positive definite, and suppose $\widehat{\Sigma}$ is such that*

$$1 - \varepsilon \le \lambda_{\min}^{1/2}(\Sigma^{-1/2}\widehat{\Sigma}\Sigma^{-1/2}) \le \lambda_{\max}^{1/2}(\Sigma^{-1/2}\widehat{\Sigma}\Sigma^{-1/2}) \le 1 + \varepsilon,$$

*where $\varepsilon \le 1/2$. Then the following inequality holds:*

$$1 - 2\varepsilon \le \lambda_{\min}^{1/2}(\Sigma^{1/2}\widehat{\Sigma}^{-1}\Sigma^{1/2}) \le \lambda_{\max}^{1/2}(\Sigma^{1/2}\widehat{\Sigma}^{-1}\Sigma^{1/2}) \le 1 + 2\varepsilon,$$

*and furthermore,*

$$\left\| \widehat{\Sigma} - \Sigma \right\|_2 \le \lambda_{\max}(\Sigma) \cdot 3\varepsilon, \tag{11}$$

$$\left\| \widehat{\Sigma}^{-1} - \Sigma^{-1} \right\|_2 \le \frac{6\varepsilon}{\lambda_{\min}(\Sigma)}, \tag{12}$$

$$\left\| \Sigma^{-1/2}\widehat{\Sigma}\Sigma^{-1/2} - I \right\|_2 \le 3\varepsilon, \tag{13}$$

$$\left\| \Sigma^{1/2}\widehat{\Sigma}^{-1}\Sigma^{1/2} - I \right\|_2 \le 6\varepsilon. \tag{14}$$

**Proof of Proposition 13.** To begin, note that the assumed inequality immediately implies that $\widehat{\Sigma}^{-1}$ is well-defined and that

$$1 - 2\varepsilon \le \lambda_{\min}^{1/2}(\Sigma^{1/2}\widehat{\Sigma}^{-1}\Sigma^{1/2}) \le \lambda_{\max}^{1/2}(\Sigma^{1/2}\widehat{\Sigma}^{-1}\Sigma^{1/2}) \le 1 + 2\varepsilon,$$

using the elementary fact that $(1 - \varepsilon)^{-1} \le 1 + 2\varepsilon$ and $(1 + \varepsilon)^{-1} \ge 1 - 2\varepsilon$ when $\varepsilon \le 1/2$. This inequality and the assumed inequality, together with Lemma 5.36 of Vershynin (2012) imply that

$$\left\| \Sigma^{-1/2}\widehat{\Sigma}\Sigma^{-1/2} - I \right\|_2 \le 3\varepsilon, \quad \text{and} \quad \left\| \Sigma^{1/2}\widehat{\Sigma}^{-1}\Sigma^{1/2} - I \right\|_2 \le 6\varepsilon.$$

Finally, observe that we can write

$$\left\| \widehat{\Sigma} - \Sigma \right\|_2 = \left\| \Sigma^{1/2}(\Sigma^{-1/2}\widehat{\Sigma}\Sigma^{-1/2} - I)\Sigma^{1/2} \right\|_2 \le \lambda_{\max}(\Sigma) \cdot \left\| \Sigma^{-1/2}\widehat{\Sigma}\Sigma^{-1/2} - I \right\|_2,$$

and

$$\left\| \widehat{\Sigma}^{-1} - \Sigma^{-1} \right\|_2 = \left\| \Sigma^{-1/2}(\Sigma^{1/2}\widehat{\Sigma}^{-1}\Sigma^{1/2} - I)\Sigma^{-1/2} \right\|_2 \le \frac{1}{\lambda_{\min}(\Sigma)} \cdot \left\| \Sigma^{1/2}\widehat{\Sigma}^{-1}\Sigma^{1/2} - I \right\|_2.$$

This establishes the result. $\qquad\square$

---

**Algorithm 4** Exp4-IX with Natarajan class and anytime guarantee

---

**Input:** policy class $\Pi$ with Natarajan dimension $d$, maximum number of rounds $T$, confidence parameter $\delta$.

**for** $k = 0, 1, \ldots$ **do**

    Explore uniformly at random for $n_k = \sqrt{2^k d \log\left(\frac{TK}{\delta}\right)}$ rounds.

    $t \leftarrow 2^k + n_k$, $\eta_k \leftarrow \sqrt{\frac{d \log\left(\frac{TK}{\delta}\right)}{2^k K}}$.

    Let $\Pi_k \in \Pi$ be the set of unique policies witnessed by $x_{2^k}, \ldots, x_{t-1}$ (choose a representative from each equivalence class).

    Initialize $P_t$ to be the uniform distribution over $\Pi_k$.

    **while** $t < 2^{k+1}$ **do**

        Receive $x_t$, sample an action $a_t \sim P_t(\cdot|x_t)$, and observe $\ell_t(a_t)$.

        Compute $P_{t+1}$ such that $P_{t+1}(\pi) \propto P_t(\pi) \exp\left(\frac{-2\eta_k\left(\frac{\ell_t(a_t)}{b}+1\right)\mathbf{1}\{a_t=\pi(x_t)\}}{P_t(a_t|x_t)+\eta_k}\right), \ \forall \pi \in \Pi_k$.

        $t \leftarrow t + 1$.

---

### C.3   Agnostic contextual bandit algorithm for Natarajan classes (Exp4-IX)

Here we present a variant of the Exp4-IX algorithm (Neu, 2015), originally proposed for achieving high-probability regret bounds for contextual bandits with a finite policy class and bounded non-negative losses. There are three main differences in the variant we present here (see Algorithm 4 for the pseudocode).

First, for our application we need an "anytime" regret guarantee that holds for any time $T' \leq T$. This can be simply achieved by a standard doubling trick. Specifically, we run the algorithm on an exponential epoch schedule (that is, epoch $k$ lasts for $2^k$ rounds) and restart Exp4-IX at the beginning of each epoch with new parameters.

Second, our policy class is infinite, but with a finite Natarajan dimension. For the two-action case with a VC class, Beygelzimer et al. (2011) gave a solution to this problem for stochastic contextual bandits. We extend their approach to Natarajan classes. Specifically, in epoch $k$ we spend the first $n_k$ rounds collecting contexts (while picking actions arbitrarily). Then we form a finite policy subset $\Pi_k$ by picking one representative from each equivalence class (that is, all policies that behave exactly the same on these contexts), and play Exp4-IX using this finite policy class $\Pi_k$ for the rest of the epoch.

Finally, in our setting losses are subgaussian and potentially unbounded. However, since with high probability they are bounded essentially by the subgaussian parameter (see Proposition 19), we simply pick $b = O\left(\tau \log^{1/2}\left(\frac{TK}{\delta}\right)\right)$ such that with probability at least $1 - \delta/3$, $\max_{a \in \mathcal{A}, t \in [T]} |\ell_t(a)| \leq b$, and transform every loss $\ell(a)$ as $\ell(a)/b + 1$, which with high probability falls in $[0, 2]$. The rest of the algorithm is the same as the original Exp4-IX. Note that we use the notation $P_t(a|x_t)$ to denote $\sum_{\pi:\pi(x_t)=a} P_t(\pi)$.

The regret guarantee for Algorithm 4 is as follows.

**Proposition 14.** *With probability at least $1 - \delta$, Algorithm 4 ensures that for all $T' \in \{1, \ldots, T\}$ and $\pi \in \Pi$,*

$$\sum_{t=1}^{T'} \ell_t(a_t) - \ell_t(\pi(x_t)) = O\left(\tau\sqrt{T'Kd}\log\left(\frac{TK}{\delta}\right)\right)$$

**Proof of Proposition 14.** We condition on the event $\max_{a \in \mathcal{A}, t \in [T]} |\ell_t(a)| \leq b$ which happens with probability at least $1 - \delta/3$. Following (Neu, 2015), with probability at least $1 - \delta/3$ we have for any $T'$ and any $\pi \in \Pi_k$ (where $k$ is the epoch containing $T'$),

$$\sum_{t=2^k}^{T'} \ell_t(a_t) - \ell_t(\pi(x_t)) = O\left(bn_k + b\sqrt{2^k K\left(\log|\Pi_k| + \log\left(\frac{TK}{\delta}\right)\right)}\right).$$

Sauer's lemma for classes with finite Natarajan dimension (Ben-David et al., 1995; Haussler and Long, 1995) gives $\log|\Pi_k| \le d\log\left(\frac{n_k e(K+1)^2}{2d}\right)$. Together with the choice of $n_k$ this gives

$$\sum_{t=2^k}^{T'} \ell_t(a_t) - \ell_t(\pi(x_t)) = O\left(b\sqrt{2^k K d \log\left(\frac{TK}{\delta}\right)}\right).$$

On the other hand, following Beygelzimer et al. (2011) we have with probability at least $1 - \delta/3$,

$$\min_{\pi \in \Pi_k} \sum_{t=2^k}^{T'} \ell_t(\pi(x_t)) \le \min_{\pi \in \Pi} \sum_{t=2^k}^{T'} \ell_t(\pi(x_t)) + O\left(\frac{b2^k d}{n_k}\log\left(\frac{TK}{\delta}\right)\right)$$

$$\le \min_{\pi \in \Pi} \sum_{t=2^k}^{T'} \ell_t(\pi(x_t)) + O\left(b\sqrt{2^k d \log\left(\frac{TK}{\delta}\right)}\right).$$

Combining these inequalities gives

$$\sum_{t=2^k}^{T'} \ell_t(a_t) \le \min_{\pi \in \Pi} \sum_{t=2^k}^{T'} \ell_t(\pi(x_t)) + O\left(b\sqrt{2^k K d \log\left(\frac{TK}{\delta}\right)}\right).$$

Summing up regret in each epoch, using $b = O\left(\tau \log^{1/2}\left(\frac{TK}{\delta}\right)\right)$, and applying a union bound leads to the result. $\qquad\square$

## C.4 Natarajan dimension for linear policy classes

**Proposition 15** (Daniely et al. (2015))**.** *Let $\phi(x,a) \in \mathbb{R}^d$ be a fixed feature map and consider the policy class*

$$\Pi = \left\{x \mapsto \operatorname*{argmax}_{a \in [K]} \langle\beta, \phi(x,a)\rangle \mid \beta \in \mathbb{R}^d\right\}.$$

*The Natarajan dimension of $\Pi$ is at most $O(d\log d)$.*

# D   Proofs from Section 3

## D.1   Square loss residual estimation

In this section we give self-contained results on estimating the square loss in a linear regression setup, extending the results of Dicker (2014) and Kong and Valiant (2018). Our main result here is the sample complexity bound for EstimateResidual described in Section 3

We first recall the abstract setting. We receive pairs $(x_1, y_1), \ldots, (x_n, y_n)$ i.i.d. from a distribution $\mathcal{D} \in \Delta(\mathbb{R}^d \times \mathbb{R})$, where $x \sim \mathsf{subG}_d(\tau^2)$ and $y \sim \mathsf{subG}(\sigma^2)$. Define $\Sigma := \mathbb{E}[xx^\top]$, and assume $\Sigma > 0$. Let $\beta^\star \in \mathbb{R}^d$ be the predictor that minimizes prediction error:

$$\beta^\star := \operatorname*{argmin}_{\beta \in \mathbb{R}^d} \mathbb{E}(\langle\beta, x\rangle - y)^2.$$

Suppose $x$ can be partitioned into features $x = (x^{(1)}, x^{(2)})$, where $x^{(1)} \in \mathbb{R}^{d_1}$ and $x^{(2)} \in \mathbb{R}^{d_2}$, and $d_1 + d_2 = d$. We define $\beta_1^\star$ to be the optimal predictor when we regress only onto the features $x^{(1)}$:

$$\beta_1^\star := \operatorname*{argmin}_{\beta \in \mathbb{R}^{d_1}} \mathbb{E}(\langle\beta, x^{(1)}\rangle - y)^2.$$

Our goal is to estimate the residual error incurred by restricting to the features $x^{(1)}$:

$$\mathcal{E} := \mathbb{E}(\langle\beta_1^\star, x^{(1)}\rangle - \langle\beta^\star, x\rangle)^2.$$

EstimateResidual (Algorithm 2) estimates $\mathcal{E}$ with sample complexity sublinear in $d$ whenever good estimates for the matrices $\Sigma$ and $\Sigma_1 := \mathbb{E}[x^{(1)}x^{(1)\top}]$ are available.[9] The performance is stated in Theorem 16. The result here is a more general version of Theorem 2, which is proven as a corollary at the end of the section.

**Theorem 16.** *Suppose the correlation matrix estimates $\widehat{\Sigma}$ and $\widehat{\Sigma}_1$ are such that*

$$1 - \varepsilon \le \lambda_{\min}^{1/2}(\Sigma^{-1/2}\widehat{\Sigma}\Sigma^{-1/2}) \le \lambda_{\max}^{1/2}(\Sigma^{-1/2}\widehat{\Sigma}\Sigma^{-1/2}) \le 1 + \varepsilon,$$

*and*

$$1 - \varepsilon \le \lambda_{\min}^{1/2}(\Sigma_1^{-1/2}\widehat{\Sigma}_1\Sigma_1^{-1/2}) \le \lambda_{\max}^{1/2}(\Sigma_1^{-1/2}\widehat{\Sigma}_1\Sigma_1^{-1/2}) \le 1 + \varepsilon,$$

*where $\varepsilon \le 1/2$. Then* EstimateResidual *guarantees that with probability at least $1 - \delta$,*

$$\left|\widehat{\mathcal{E}} - \mathcal{E}\right| \le \frac{1}{2}\mathcal{E} + O\left(\frac{\lambda_{\max}(\Sigma)}{\lambda_{\min}^2(\Sigma)} \cdot \frac{\sigma^2\tau^2 d^{1/2}\log^2(2d/\delta)}{n} + \frac{\lambda_{\max}(\Sigma)}{\lambda_{\min}^2(\Sigma)}\|\mathbb{E}[xy]\|_2^2 \cdot \varepsilon^2\right). \tag{15}$$

**Proof of Theorem 16.** We begin by giving an expression for $\mathcal{E}$ that will make the choice of estimator more transparent. Let

$$D = \begin{pmatrix} \Sigma_1 & 0_{d_1 \times d_2} \\ 0_{d_2 \times d_1} & 0_{d_2 \times d_2} \end{pmatrix},$$

and let $R = D^\dagger - \Sigma^{-1}$. Observe that by first-order conditions, we may take $\beta^\star = \Sigma^{-1}\mathbb{E}[xy]$ and $\beta_1^\star = \Sigma_1^{-1}\mathbb{E}[x^{(1)}y]$. Moreover, for any $x$, we may write $\langle \beta_1^\star, x^{(1)} \rangle = \langle D^\dagger \mathbb{E}[xy], x \rangle$. Consequently, we have

$$\mathcal{E} = \mathbb{E}\left(\langle D^\dagger \mathbb{E}[xy], x \rangle - \langle \Sigma^{-1}\mathbb{E}[xy], x \rangle\right)^2 = \mathbb{E}\langle R\mathbb{E}[xy], x \rangle^2 = \left\|\Sigma^{1/2}R\mathbb{E}[xy]\right\|_2^2.$$

With this representation, it is clear that if $\widehat{\Sigma} = \Sigma$ and $\widehat{\Sigma}_1 = \Sigma_1$, then $\widehat{\mathcal{E}}$ is an unbiased estimator for $\mathcal{E}$. Our proof has two parts: We first show that $\widehat{\mathcal{E}}$ concentrates around its expectation, then bound the bias due to $\widehat{\Sigma}$ and $\widehat{\Sigma}_1$.

For concentration, we appeal to Lemma 17. Note that $\lambda_{\min}(\Sigma_1) \ge \lambda_{\min}(\Sigma)$ and $\lambda_{\max}(\Sigma_1) \le \lambda_{\max}(\Sigma)$; this can be seen using the variational representation for the eigenvalues. Consequently by Proposition 13 we have that

$$\lambda_{\max}(\widehat{\Sigma}) \vee \lambda_{\max}(\widehat{\Sigma}_1) \le O(\lambda_{\max}(\Sigma)), \quad \text{and,} \quad \lambda_{\min}(\widehat{\Sigma}) \wedge \lambda_{\min}(\widehat{\Sigma}_1) \ge \Omega(\lambda_{\min}(\Sigma)).$$

This implies that

$$\left\|\widehat{\Sigma}^{1/2}\widehat{R}\right\|_2 \le O\left(\lambda_{\max}^{1/2}(\Sigma)/\lambda_{\min}(\Sigma)\right),$$

and so by Proposition 9, it follows that the random variable $\widehat{\Sigma}^{1/2}\widehat{R}xy - \mathbb{E}[\widehat{\Sigma}^{1/2}\widehat{R}xy]$ has subexponential parameter of order $O(\sigma\tau\lambda_{\max}^{1/2}(\Sigma)/\lambda_{\min}(\Sigma))$. Thus, by Lemma 17, we have that with probability at least $1 - \delta$,

$$\left|\widehat{\mathcal{E}} - \mathbb{E}[\widehat{\mathcal{E}}]\right| \le O\left(\frac{\lambda_{\max}(\Sigma)}{\lambda_{\min}^2(\Sigma)}\frac{\sigma^2\tau^2 d^{1/2}\log^2(2d/\delta)}{n} + \frac{\lambda_{\max}^{1/2}(\Sigma)}{\lambda_{\min}(\Sigma)}\frac{\sigma\tau\|\widehat{\Sigma}^{1/2}\widehat{R}\mathbb{E}[xy]\|_2\log(2/\delta)}{\sqrt{n}}\right).$$

But note that since $\|\widehat{\Sigma}^{1/2}\widehat{R}\mathbb{E}[xy]\|_2 = \sqrt{\mathbb{E}[\widehat{\mathcal{E}}]}$, we can apply the AM-GM inequality to deduce

$$\left|\widehat{\mathcal{E}} - \mathbb{E}[\widehat{\mathcal{E}}]\right| \le \frac{1}{8}\mathbb{E}[\widehat{\mathcal{E}}] + O\left(\frac{\lambda_{\max}(\Sigma)}{\lambda_{\min}^2(\Sigma)} \cdot \frac{\sigma^2\tau^2 d^{1/2}\log^2(2d/\delta)}{n}\right).$$

We now bound the error from $\mathbb{E}[\widehat{\mathcal{E}}]$ to $\mathcal{E}$. With the shorthand $\mu = \mathbb{E}[xy]$, observe that we have

$$\mathbb{E}[\widehat{\mathcal{E}}] - \mathcal{E} = \langle \widehat{R}\widehat{\Sigma}\widehat{R}\mu, \mu \rangle - \langle R\Sigma R\mu, \mu \rangle$$
$$= 2\langle (\widehat{R} - R)\widehat{\Sigma}R\mu, \mu \rangle + \langle R(\widehat{\Sigma} - \Sigma)R\mu, \mu \rangle + \langle (\widehat{R} - R)\widehat{\Sigma}(\widehat{R} - R)\mu, \mu \rangle.$$

Applying Cauchy-Schwarz to each term, we get an upper bound of

$$\left|\mathbb{E}[\widehat{\mathcal{E}}] - \mathcal{E}\right| \le 2\left\|\Sigma^{-1/2}\widehat{\Sigma}(R - \widehat{R})\mu\right\|_2\left\|\Sigma^{1/2}R\mu\right\|_2 + \left\|\Sigma^{-1/2}(\Sigma - \widehat{\Sigma})R\mu\right\|_2\left\|\Sigma^{1/2}R\mu\right\|_2$$
$$+ \left\|\widehat{\Sigma}^{1/2}(R - \widehat{R})\mu\right\|_2^2.$$

Since $\left\|\Sigma^{1/2}R\mu\right\|_2 = \sqrt{\mathcal{E}}$, we can apply the AM-GM inequality to each of the first two terms to conclude that

$$\left|\mathbb{E}[\widehat{\mathcal{E}}] - \mathcal{E}\right| \le \frac{1}{8}\mathcal{E} + O\left(\left\|\Sigma^{-1/2}\widehat{\Sigma}(R - \widehat{R})\mu\right\|_2^2 + \left\|\Sigma^{-1/2}(\Sigma - \widehat{\Sigma})\widehat{R}\mu\right\|_2^2 + \left\|\widehat{\Sigma}^{1/2}(R - \widehat{R})\mu\right\|_2^2\right). \tag{16}$$

To proceed, we first collect a number of spectral properties, all of which follow from the assumptions in the theorem statement and Proposition 13:

$$\left\|\Sigma^{-1/2}\widehat{\Sigma}\right\|_2 = \left\|\Sigma^{-1/2}\widehat{\Sigma}\Sigma^{-1/2}\Sigma^{1/2}\right\|_2 \le (1+\varepsilon)\left\|\Sigma^{1/2}\right\|_2 \le O(\lambda_{\max}^{1/2}(\Sigma)),$$

$$\left\|\widehat{R}\right\|_2 \le O(1/\lambda_{\min}(\Sigma)),$$

$$\left\|R - \widehat{R}\right\|_2 \le \left\|\widehat{\Sigma}_1^{-1} - \Sigma_1^{-1}\right\|_2 + \left\|\widehat{\Sigma}^{-1} - \Sigma^{-1}\right\|_2 \le O(\varepsilon/\lambda_{\min}(\Sigma)),$$

$$\left\|\Sigma^{-1/2}\widehat{\Sigma}\Sigma^{-1/2} - I\right\|_2 \le O(\varepsilon).$$

We now bound the terms in (16) one by one. Using the spectral bounds above, we have

$$\left\|\Sigma^{-1/2}\widehat{\Sigma}(R-\widehat{R})\mu\right\|_2^2 \le \left\|\Sigma^{-1/2}\widehat{\Sigma}\right\|_2^2 \|\mu\|_2^2 \cdot \left\|R-\widehat{R}\right\|_2^2 \le O\left(\frac{\lambda_{\max}(\Sigma)}{\lambda_{\min}^2(\Sigma)}\|\mu\|_2^2 \cdot \varepsilon^2\right),$$

$$\left\|\Sigma^{-1/2}(\Sigma-\widehat{\Sigma})\widehat{R}\mu\right\|_2^2 \le \|\Sigma\|_2 \left\|\widehat{R}\right\|_2^2 \|\mu\|_2^2 \cdot \left\|\Sigma^{-1/2}\widehat{\Sigma}\Sigma^{-1/2} - I\right\|_2^2 \le O\left(\frac{\lambda_{\max}(\Sigma)}{\lambda_{\min}^2(\Sigma)}\|\mu\|_2^2 \cdot \varepsilon^2\right),$$

$$\left\|\widehat{\Sigma}^{1/2}(R-\widehat{R})\mu\right\|_2^2 \le \left\|\widehat{\Sigma}\right\|_2 \|\mu\|_2^2 \cdot \left\|R-\widehat{R}\right\|_2^2 \le O\left(\frac{\lambda_{\max}(\Sigma)}{\lambda_{\min}^2(\Sigma)}\|\mu\|_2^2 \cdot \varepsilon^2\right).$$

We conclude that

$$\left|\mathbb{E}[\widehat{\mathcal{E}}] - \mathcal{E}\right| \le \frac{1}{8}\mathcal{E} + O\left(\frac{\lambda_{\max}(\Sigma)}{\lambda_{\min}^2(\Sigma)}\|\mu\|_2^2 \cdot \varepsilon^2\right),$$

which yields the result. $\qquad\square$

**Lemma 17.** *Let $z$ be a random variable such that $z - \mathbb{E}[z] \sim \mathsf{subE}_d(\lambda)$, and let $z_1, \ldots, z_n$ be i.i.d. copies. Define*

$$W = \sum_{i<j}\langle z_i, z_j\rangle.$$

*Then with probability at least $1 - \delta$,*

$$|W - \mathbb{E}[W]| \le O\left(\lambda^2 d^{1/2} n \log^2(2d/\delta) + \lambda n^{3/2}\|\mathbb{E}[z]\|_2 \log(2/\delta)\right).$$

**Proof of Lemma 17.** First observe that we can write

$$|W - \mathbb{E}[W]| = \left|\sum_{i<j}\langle z_i, z_j\rangle - \langle\mathbb{E}[z], \mathbb{E}[z]\rangle\right|$$

$$= \left|\sum_{i<j}\langle z_i - \mathbb{E}[z], z_j - \mathbb{E}[z]\rangle + \sum_{i<j}\langle z_i - \mathbb{E}[z], \mathbb{E}[z]\rangle + \sum_{i<j}\langle z_j - \mathbb{E}[z], \mathbb{E}[z]\rangle\right|$$

$$\le \underbrace{\left|\sum_{i<j}\langle z_i - \mathbb{E}[z], z_j - \mathbb{E}[z]\rangle\right|}_{S_1} + \underbrace{(n-1)\left|\sum_{i=1}^n\langle z_i - \mathbb{E}[z], \mathbb{E}[z]\rangle\right|}_{S_2}.$$

We bound $S_2$ first. Define $B = \|\mathbb{E}[z]\|_2$. Observe that for each $i$, the summand is subexponential: $\langle z_i - \mathbb{E}[z], \mathbb{E}[z]\rangle \sim \mathsf{subE}(\lambda B)$. Consequently, Bernstein's inequality implies that with probability at least $1 - \delta$,

$$S_2 \le O\left(\lambda Bn^{3/2}(\log(2/\delta))^{1/2} + 2\lambda Bn\log(2/\delta)\right).$$

For $S_1$, we first apply a decoupling inequality. Let $z_1', \ldots, z_n'$ be a sequence of independent copies of $z_1, \ldots, z_n$. Then by Theorem 3.4.1 of de la Peña and Giné (1998), there are universal constants $C, c > 0$ such that for any $t \ge 0$,

$$\Pr(S_1 > t) \le C\Pr\left(\left|\sum_{i<j}\langle z_i - \mathbb{E}[z], z_j' - \mathbb{E}[z]\rangle\right| > t/c\right).$$

We write

$$\sum_{i<j}\langle z_i - \mathbb{E}[z], z_j' - \mathbb{E}[z]\rangle = \sum_{i=1}^n\left\langle z_i - \mathbb{E}[z], \sum_{j=i+1}^n z_j' - \mathbb{E}[z]\right\rangle = \sum_{i=1}^n\langle z_i - \mathbb{E}[z], Z_i\rangle,$$

where $Z_i = \sum_{j=i+1}^n z_j' - \mathbb{E}[z]$. Now condition on $z_1', \ldots, z_n'$. Then $\langle z_i - \mathbb{E}[z], Z_i \rangle \sim \mathsf{subE}(\lambda \|Z_i\|_2)$. Thus, by Bernstein's inequality, we have that with probability at least $1 - \delta$, over the draw of $z_1, \ldots, z_n$,

$$\left| \sum_{i<j} \langle z_i - \mathbb{E}[z], z_j' - \mathbb{E}[z] \rangle \right| \le O\Big( \lambda (n \log(2/\delta))^{1/2} + \lambda \log(2/\delta) \Big) \cdot \max_{i \in [n]} \|Z_i\|_2.$$

Next, using Lemma 11, we have that with probability at least $1 - \delta$,

$$\max_{i \in [n]} \|Z_i\|_2 \le O\Big( \lambda \sqrt{dn \log(2d/\delta)} + \lambda d^{1/2} \log(2d/\delta) \Big).$$

Thus, by union bound, after grouping terms we get that with probability at least $1 - \delta$,

$$\left| \sum_{i<j} \langle z_i - \mathbb{E}[z], z_j' - \mathbb{E}[z] \rangle \right| \le O\big( \lambda^2 d^{1/2} n \log^2(2d/\delta) \big),$$

and

$$S_2 \le O\big( \lambda^2 d^{1/2} n \log^2(2d/\delta) \big).$$

Taking a union bound yields the final result. □

**Proof of Theorem 2.** Consider the distribution over random vectors $x' := \Sigma^{-1/2} x$, and let $\widehat{\Sigma}'$ denote the empirical covariance under this distribution. Since $\mathbb{E}[x' x'^\top] = I$, we may apply Proposition 12, which implies that with probability at least $1 - \delta$,

$$1 - \varepsilon \le \lambda_{\min}^{1/2}(\widehat{\Sigma}') \le \lambda_{\max}^{1/2}(\widehat{\Sigma}') \le 1 + \varepsilon,$$

where $\varepsilon \le 50 \lambda_{\min}^{-1}(\Sigma) \tau^2 \sqrt{\frac{d + \log(2/\delta)}{m}}$, and we have used that the subgaussian parameter of $x'$ is at most $\lambda_{\min}^{-1}(\Sigma)$ times that of $x$. We can equivalently write this expression as

$$1 - \varepsilon \le \lambda_{\min}^{1/2}(\Sigma^{-1/2} \widehat{\Sigma} \Sigma^{-1/2}) \le \lambda_{\max}^{1/2}(\Sigma^{-1/2} \widehat{\Sigma} \Sigma^{-1/2}) \le 1 + \varepsilon.$$

Note also that once $m \ge C(d + \log(2/\delta)) \tau^4 / \lambda_{\min}(\Sigma)$ for some universal constant $C$, we have $\varepsilon \le 1/2$. Applying the same reasoning to $\widehat{\Sigma}_1$ and taking a union bound, then appealing to Theorem 16 yields the result. We use that $\lambda_{\max}(\Sigma) \le \tau^2$ to simplify the final expression. □

## D.2 Proof of Theorem 3

### D.2.1 Basic technical results

In this section we prove some utility results that bound the scale of various random variables appearing in the analysis of ModCB.

**Proposition 18.** *For all $a$, $\ell(a) \sim \mathsf{subG}(4\tau^2)$.*

**Proof.** We have $\ell(a) = \langle \beta^\star, \phi^{m^\star}(x) \rangle + \varepsilon_a$, where $\varepsilon_a = \ell(a) - \mathbb{E}[\ell(a) \mid x]$. Note that $\langle \beta^\star, \phi^{m^\star}(x) \rangle$ has subgaussian parameter $B^2 \tau^2$, and $\varepsilon_a$ has subgaussian parameter $\sigma^2$, so the triangle inequality for subgaussian parameters implies that the parameter of $\ell(a)$ is at most $(B\tau + \sigma)^2 \le 4\tau^2$, where we have used the assumption that $B \le 1$ and $\sigma \le \tau$. □

**Proposition 19.** *With probability at least $1 - \delta$, $\max_{a \in \mathcal{A}, t \in [T]} |\ell_t(a)| \le O\Big( \tau \sqrt{\log(KT/\delta)} \Big)$.*

**Proof.** Immediate consequence of Proposition 18, along with Hoeffding's inequality and a union bound. □

### D.2.2 Square loss translation

We first introduce some additional notation. Let $L_m^\star = \min_{\pi \in \Pi_m} L(\pi)$. Recall that

$$\beta_m^\star = \operatorname*{argmin}_{\beta \in \mathbb{R}^{d_m}} \frac{1}{K} \sum_{a \in \mathcal{A}} \mathbb{E}_{x \sim \mathcal{D}} (\langle \beta, \phi^m(x, a) \rangle - \ell(a))^2.$$

We let $\pi_m^{\mathrm{sq}}(x) = \operatorname*{argmin}_{a \in \mathcal{A}} \langle \beta_m^\star, \phi^m(x, a) \rangle$ be the induced policy.

**Proposition 20.** *For all $m \in [M]$, $\beta_m^\star$ is uniquely defined, and $\beta_m^\star = (\beta^\star, \mathbf{0}_{d_m - d_{m^\star}})$ for all $m \geq m^\star$. As a consequence:*

- $\pi_m^{\mathrm{sq}} = \pi^\star$ *for all $m \geq m^\star$.*
- $\mathcal{E}_{i,m^\star} = \mathcal{E}_{i,j}$ *for all $i \in [M]$ and all $j \geq m^\star$.*
- $\mathcal{E}_{i,j} = 0$ *for all $j > i \geq m^\star$.*

**Proof of Proposition 20.** That each $\beta_m^\star$ is uniquely defined follows from the assumption that $\lambda_{\min}(\Sigma_m) > 0$, since this implies that the optimization problem (3) is strongly convex.

To show that $\beta_m^\star = (\beta^\star, \mathbf{0}_{d_m - d_{m^\star}})$ when $m \geq m^\star$, observe that by first order conditions, $\beta_m^\star$ is uniquely defined via

$$\beta_m^\star = \Sigma_m^{-1} \cdot \frac{1}{K} \sum_{a \in \mathcal{A}} \mathbb{E}_{(x,\ell) \sim \mathcal{D}}[\phi^m(x, a)\ell(a)].$$

But note that for each $a \in \mathcal{A}$, the realizability assumption (1) implies that

$$\mathbb{E}_{(x,\ell) \sim \mathcal{D}}[\phi^m(x, a)\ell(a)] = \mathbb{E}_{x \sim \mathcal{D}}\Big[\phi^m(x, a)\langle \phi^{m^\star}(x, a), \beta^\star \rangle\Big]$$

$$= \mathbb{E}_{x \sim \mathcal{D}}\Big[\phi^m(x, a)\phi^m(x, a)^\top (\beta^\star, \mathbf{0}_{d_m - d_{m^\star}})\Big],$$

where the last equality follows from the nested feature map assumption. Combining this with the preceding identity, we have

$$\beta_m^\star = \Sigma_m^{-1} \Sigma_m (\beta^\star, \mathbf{0}_{d_m - d_{m^\star}}) = (\beta^\star, \mathbf{0}_{d_m - d_{m^\star}}).$$

The remaining claims now follow immediately from the definition of $\pi^{\mathrm{sq}}$ and $\mathcal{E}_{i,j}$. $\square$

**Proposition 21.** *For all $a \in \mathcal{A}$ and $m \in [M]$, we have $\|\mathbb{E}[\phi^m(x, a)\ell(a)]\|_2 \leq \tau^2$. For all $m \in [M]$, we have $\|\beta_m^\star\|_2 \leq \tau/\gamma$, and so $\pi_m^{\mathrm{sq}} \in \Pi_m$.*

**Proof.** For the first claim, we use realizability to write

$$\mathbb{E}[\phi^m(x, a)\ell(a)] = \mathbb{E}\Big[\phi^m(x, a)\langle \phi^{m^\star}(x, a), \beta^\star \rangle\Big]$$

Hence any $\theta \in \mathbb{R}^{d_m}$ with $\|\theta\|_2 \leq 1$, we have

$$\langle \mathbb{E}[\phi^m(x, a)\ell(a)], \theta \rangle = \mathbb{E}\Big[\langle \phi^m(x, a), \theta \rangle \langle \phi^{m^\star}(x, a), \beta^\star \rangle\Big]$$

$$\leq \sqrt{\mathbb{E}\langle \phi^m(x, a), \theta \rangle^2 \cdot \mathbb{E}\langle \phi^{m^\star}(x, a), \beta^\star \rangle^2}$$

$$\leq \tau^2,$$

where the first inequality is Cauchy-Schwarz and the second follows because $\|\beta^\star\|_2 \leq 1$ by assumption and all feature maps belong to $\mathsf{subG}_{d_m}(\tau^2)$. For the second claim, recall as in the proof of Proposition 20 that $\beta_m^\star$ is uniquely defined as

$$\beta_m^\star = \Sigma_m^{-1} \cdot \frac{1}{K} \sum_{a \in \mathcal{A}} \mathbb{E}_{(x,\ell) \sim \mathcal{D}}[\phi^m(x, a)\ell(a)] = \frac{1}{K} \sum_{a \in \mathcal{A}} \mathbb{E}_{(x,\ell) \sim \mathcal{D}}\Big[\Sigma_m^{-1}\phi^m(x, a)\langle \phi^{m^\star}(x, a), \beta^\star \rangle\Big].$$

Following the same approach as for the first claim, for any any $\theta \in \mathbb{R}^{d_m}$ with $\|\theta\|_2 \leq 1$, we have

$$\langle \theta, \beta_m^\star \rangle = \frac{1}{K} \sum_{a \in \mathcal{A}} \mathbb{E}_{(x,\ell) \sim \mathcal{D}}\Big[\langle \Sigma_m^{-1}\phi^m(x, a), \theta \rangle \langle \phi^{m^\star}(x, a), \beta^\star \rangle\Big]$$

$$\leq \sqrt{\frac{1}{K} \sum_{a \in \mathcal{A}} \mathbb{E}_{(x,\ell) \sim \mathcal{D}}\langle \Sigma_m^{-1}\phi^m(x, a), \theta \rangle^2 \cdot \frac{1}{K} \sum_{a \in \mathcal{A}} \mathbb{E}_{(x,\ell) \sim \mathcal{D}}\langle \phi^{m^\star}(x, a), \beta^\star \rangle^2}$$

$$= \sqrt{\langle \Sigma_m^{-1}\theta, \theta \rangle \cdot \frac{1}{K} \sum_{a \in \mathcal{A}} \mathbb{E}_{(x,\ell) \sim \mathcal{D}}\langle \phi^{m^\star}(x, a), \beta^\star \rangle^2}$$

$$\leq \tau/\gamma,$$

where we again have used that $\|\beta^\star\|_2 \leq 1$. $\square$

**Lemma 22.** *For all $i \in [M]$, and all $j \geq m^\star$,*

$$\Delta_{i,j} \leq L(\pi_i^{\mathrm{sq}}) - L(\pi_j^{\mathrm{sq}}) \leq \sqrt{4K \cdot \mathcal{E}_{i,j}}. \tag{17}$$

**Proof of Lemma 22.** Observe that we have

$$L_i^\star - L_j^\star \leq L(\pi_i^{\mathrm{sq}}) - L(\pi_j^\star) = L(\pi_i^{\mathrm{sq}}) - L(\pi_j^{\mathrm{sq}}),$$

where the inequality holds because $\pi_i^{\mathrm{sq}} \in \Pi_i$ and the equality follows from Proposition 20 and the assumption that $j \geq m^\star$. Using realizability along with the representation for $\beta_j^\star$ from Proposition 20, we write

$$L(\pi_i^{\mathrm{sq}}) - L(\pi_j^{\mathrm{sq}}) = \mathbb{E}_{x \sim \mathcal{D}}\left[\left\langle \beta_j^\star, \phi^j(x, \pi_i^{\mathrm{sq}}(x))\right\rangle - \left\langle \beta_j^\star, \phi^j(x, \pi_j^{\mathrm{sq}}(x))\right\rangle\right]$$

For each $x$, we have

$$\left\langle \beta_i^\star, \phi^i(x, \pi_j^{\mathrm{sq}}(x))\right\rangle - \left\langle \beta_i^\star, \phi^i(x, \pi_i^{\mathrm{sq}}(x))\right\rangle \geq 0,$$

which follows from the definition of $\pi_i^{\mathrm{sq}}$. We add this inequality to the preceding equation to get

$$\begin{aligned}
L(\pi_i^{\mathrm{sq}}) - L(\pi_j^{\mathrm{sq}}) &\leq \mathbb{E}_{x \sim \mathcal{D}}\left[\left\langle \beta_j^\star, \phi^j(x, \pi_i^{\mathrm{sq}}(x))\right\rangle - \left\langle \beta_i^\star, \phi^i(x, \pi_i^{\mathrm{sq}}(x))\right\rangle\right] \\
&\quad + \mathbb{E}_{x \sim \mathcal{D}}\left[\left\langle \beta_i^\star, \phi^i(x, \pi_j^{\mathrm{sq}}(x))\right\rangle - \left\langle \beta_j^\star, \phi^j(x, \pi_j^{\mathrm{sq}}(x))\right\rangle\right] \\
&\leq 2\, \mathbb{E}_{x \sim \mathcal{D}} \max_a \left|\left\langle \beta_i^\star, \phi^i(x, a)\right\rangle - \left\langle \beta_j^\star, \phi^j(x, a)\right\rangle\right|.
\end{aligned}$$

Lastly, using Jensen's inequality, we have

$$\begin{aligned}
\mathbb{E}_{x \sim \mathcal{D}} \max_a \left|\left\langle \beta_i^\star, \phi^i(x, a)\right\rangle - \left\langle \beta_j^\star, \phi^j(x, a)\right\rangle\right| &\leq \sqrt{\mathbb{E}_{x \sim \mathcal{D}} \max_a \left(\left\langle \beta_i^\star, \phi^i(x, a)\right\rangle - \left\langle \beta_j^\star, \phi^j(x, a)\right\rangle\right)^2} \\
&\leq \sqrt{K \cdot \tfrac{1}{K} \sum_{a \in \mathcal{A}} \mathbb{E}_{x \sim \mathcal{D}} \left(\left\langle \beta_i^\star, \phi^i(x, a)\right\rangle - \left\langle \beta_j^\star, \phi^j(x, a)\right\rangle\right)^2} \\
&= \sqrt{K \cdot \mathcal{E}_{i,j}}. \qquad \square
\end{aligned}$$

### D.2.3 Decomposition of regret

To proceed with the analysis we require some additional notation. We let $N$ be the number of values that $\widehat{m}$ takes on throughout the execution of the algorithm (i.e., $N$ is the number of candidate policy classes that are tried), and let $\widehat{m}_k$ for $k \in [N]$ denote the $k$th such value. We let $\mathcal{I}_k \subseteq [T]$ denote the interval for which $\widehat{m} = \widehat{m}_k$, and let $T_k$ denote the first timestep in this interval.

We let $S_t$ denote the value of the set $S$ at step $t$ (after uniform exploration has occurred, if it occurred). We let $\overline{\mathcal{I}}_k = \mathcal{I}_k \setminus S_T$ denote the rounds in interval $k$ in which uniform exploration did not occur.

We let $\widehat{\mathcal{E}}_{ij}(t)$ the random variable defined by running EstimateResidual using the dataset $H_j(t) = \left\{(\phi^j(x_s, a_s), \ell_s(a_s))\right\}_{s \in S_t}$ and empirical second moment matrices $\widehat{\Sigma}_i$ and $\widehat{\Sigma}_j$ at time $t$. Note that $\widehat{\mathcal{E}}_{i,j}(t)$ is well-defined even for pairs $(i, j)$ for which the algorithm does not invoke EstimateResidual at time $t$.

We partition the intervals as follows: Let $k_0$ be the first interval containing $t \geq T_{m^\star}^{\min}$, and let $k_1$ be the first inteval for which $k \geq m^\star$. We will eventually show that with high probability $k_1 \geq N$, or in other words, once the algorithm reaches a class containing the optimal policy it never leaves (if it reaches such a class, that is).

Let $\mathrm{Reg}(\mathcal{I}_k)$ denote the regret to $m^\star$ incurred throughout interval $k$. We bound regret to $\pi^\star$ as

$$\mathrm{Reg} \leq \sum_{k=1}^{k_0-1} \mathrm{Reg}(\mathcal{I}_k) + \sum_{k=k_0}^{k_1-1} \mathrm{Reg}(\mathcal{I}_k) + \sum_{k=k_1}^{N} \mathrm{Reg}(\mathcal{I}_k). \tag{18}$$

The main result in this subsection is Lemma 23 which shows that with high probability, the estimators $\widehat{\mathcal{E}}_{i,j}$ and Exp4-IX instances invoked by the algorithm behave as expected, and various quantities arising in the regret analysis are bounded appropriately.

**Lemma 23.** *Let $A$ be the event that the following properties hold:*

*1. For all $a \in \mathcal{A}$ and $t \in [T]$, $|\ell_t(a)| \leq O\left(\tau\sqrt{\log(KT/\delta_0)}\right)$.*         (event $A_1$)

2. *For all $t \geq T_1^{\min}$, $\frac{1}{8}K^\kappa t^{1-\kappa} \leq |S_t| \leq 4K^\kappa t^{1-\kappa}$.* (event $A_2$)

3. *For all $i < j$, for all $t \geq T_j^{\min}$, $\left|\widehat{\mathcal{E}}_{i,j}(t) - \mathcal{E}_{i,j}\right| \leq \frac{1}{2}\mathcal{E}_{i,j} + \alpha_{j,t}$.* (event $A_3$)

4. *For all $k$, $\sum_{t \in \overline{\mathcal{I}}_k} \ell_t(a_t) - \ell_t(\pi_{\widehat{m}_k}^{\mathrm{sq}}(x_t)) \leq O\left(\tau\sqrt{d_{\widehat{m}_k}|\overline{\mathcal{I}}_k| \cdot K \log^2(TK/\delta_0) \log d_{\widehat{m}_k}}\right)$.* (event $A_4$)

5. *For all $i, j \in [M]$ and all intervals $\mathcal{I} = [t_1, t_2]$,*
$$\sum_{t \in \mathcal{I}} \ell_t(\pi_i^{\mathrm{sq}}(x_t)) - \ell_t(\pi_j^{\mathrm{sq}}(x_t)) \leq |\mathcal{I}| \cdot (L(\pi_i^{\mathrm{sq}}) - L(\pi_j^{\mathrm{sq}})) + O\left(\tau\sqrt{|\mathcal{I}|\log(2/\delta_0)}\right).$$
(event $A_5$)

*When $C_1$ and $C_2$ are sufficiently large constants, event $A$ holds with probability at least $1 - \delta$.*

**Proof of Lemma 23.** First, note that event $A_1$ holds with probability at least $1 - \delta/10$ by Proposition 19.

We now move on to $A_2$. For any fixed $t$, Bernstein's inequality implies that with probability at least $1 - \delta_0$,
$$|S_t| \geq \mathbb{E}_t|S_t| - \sqrt{4\,\mathbb{E}|S_t|\log(2/\delta_0)} - \log(2/\delta_0) \geq \frac{1}{2}\,\mathbb{E}|S_k| - 3\log(2/\delta_0),$$
and likewise implies that $|S_t| \leq \frac{3}{2}\,\mathbb{E}|S_t| + 3\log(2/\delta_0)$. Next, note that
$$\mathbb{E}|S_t| = \sum_{s=1}^{t}\left(1 \wedge \frac{K^\kappa}{s^\kappa}\right).$$

It follows that $\mathbb{E}|S_t| \leq 2K^\kappa t^{1-\kappa}$. We also have $\mathbb{E}|S_t| \geq K^\kappa(t^{1-\kappa} - 2K^{1-\kappa})$, which is lower bounded by $K^\kappa t^{1-\kappa}/2$ once $t \geq 4^{\frac{1}{1-\kappa}}K$, and in particular once $t \geq T_1^{\min}$ whenever $C_2$ is sufficiently large. If we union bound over all $t$, these results together imply that once $t \geq (30\log(2/\delta_0)/K^\kappa)^{\frac{1}{1-\kappa}}$ (which is implied by $t \geq T_1^{\min}$ when $C_2$ is large enough), then with probability at least $1 - \delta_0 T \geq 1 - \delta/10$, for all $t$,
$$\frac{1}{8}K^\kappa t^{1-\kappa} \leq |S_t| \leq 4K^\kappa t^{1-\kappa}.$$

For $A_3$, let $t$ and $i < j$ be fixed. Note that conditioned on the size of $|S_t|$, the examples $\left\{\phi^j(x_s, a_s), \ell_s(a_s)\right\}_{s \in S_t}$ are i.i.d. Consequently, Theorem 2 implies that with probability at least $1 - \delta_0$,
$$\left|\widehat{\mathcal{E}}_{i,j}(t) - \mathcal{E}_{i,j}\right| \leq \frac{1}{2}\mathcal{E}_{i,j} + O\left(\frac{\tau^6}{\gamma^4} \cdot \frac{d_j^{1/2}\log^2(2d_j/\delta_0)}{|S_t|} + \frac{\tau^{10}}{\gamma^8} \cdot \frac{d_j\log(2/\delta_0)}{t}\right),$$
where we have used Proposition 18 to show that $\ell_s(a_s) \sim \mathsf{subG}(4\tau^2)$ and used Proposition 21 to show that $\left\|\frac{1}{K}\sum_{a \in \mathcal{A}}\mathbb{E}[\phi^j(x,a)\ell(a)]\right\|_2 \leq \tau^2$. Conditioned on $A_2$, we have $|S_t| = \Omega(K^\kappa t^{1-\kappa})$ once $t \geq T_1^{\min}$, and so when $C_1$ is a sufficiently large absolute constant, $\left|\widehat{\mathcal{E}}_{i,j} - \mathcal{E}_{i,j}\right| \leq \frac{1}{2}\mathcal{E}_{i,j} + \alpha_{j,t}$. By union bound, we get that conditioned on event $A_2$, event $A_3$ holds with probability at least $1 - M^3 T\delta_0 \geq 1 - \delta/10$.

To prove $A_4$, we appeal to Proposition 14. To do so, we verify the following facts: 1. Losses belong to $\mathsf{subG}(4\tau^2)$ (Proposition 18) 2. Each policy class $\Pi_m$ is compact and contains $\pi_m^{\mathrm{sq}}$ (compactness is immediate, containment of $\pi_m^{\mathrm{sq}}$ follows from Proposition 21) 3. The Natarajan dimension of $\Pi_m$ is at most $O(d_m \log(d_m))$ (Proposition 15).

Now let $k \in [N]$ be fixed. Since Proposition 14 provides an anytime regret guarantee for Exp4-IX, and since the context-loss pairs fed into the algorithm still follow the distribution $\mathcal{D}$ (the step at which we perform uniform exploration does not alter the distribution). Consequently, conditioned on the history up until time $T_k$, we have that with probability at least $1 - \delta_0$,
$$\sum_{t \in \overline{\mathcal{I}}_k} \ell_t(a_t) - \ell_t(\pi_{\widehat{m}_k}^{\mathrm{sq}}(x_t)) \leq O\left(\tau\sqrt{d_{\widehat{m}_k}|\overline{\mathcal{I}}_k| \cdot K \log^2(TK/\delta_0) \log d_{\widehat{m}_k}}\right),$$

since $\overline{\mathcal{I}}_k$ is precisely the set of rounds for which the Exp4-IX instance was active in epoch $k$. Taking a union bound over all $M$ Exp4-IX instances and all possible starting times for each instances, we have that with probability at least $1 - MT\delta_0 \geq 1 - \delta/10$, the inequality above holds for all $k$.

For $A_5$, let $i, j \in [M]$ and the interval $\mathcal{I} = \{t_1, \ldots, t_2\} \subset [T]$ be fixed. The, since $\ell_t(a) \sim \mathsf{subG}(4\tau^2)$ for all $a$. Hoeffding's inequality implies that with probability at least $1 - \delta_0$,

$$\sum_{t \in \mathcal{I}} \ell_t(\pi_i^{\mathrm{sq}}(x_t)) - \ell_t(\pi_j^{\mathrm{sq}}(x_t)) \leq |\mathcal{I}| \cdot (L(\pi_i^{\mathrm{sq}}) - L(\pi_j^{\mathrm{sq}})) + O\left(\tau\sqrt{|\mathcal{I}|\log(2/\delta_0)}\right).$$

By a union bound over all $i, j$ pairs and all such intervals, we have that $A_5$ occurs with probability at least $1 - M^2 T^2 \delta_0 \geq 1 - \delta/10$. Taking a union bound over events $A_1$ through $A_5$ leads to the final result. $\qquad \square$

### D.2.4 Final bound

We now use the regret decomposition (18) in conjunction with Lemma 23 to prove the theorem. We use $\tilde{O}$ to suppress factors logarithmic in $K$, $T$, $M$, and $\log(1/\delta)$.

Condition on the event $A$ in Lemma 23, which happens with probability at least $1 - \delta$ so long as $C_1$ and $C_2$ are sufficiently large absolute constants.

We begin from the regret decomposition

$$\mathrm{Reg} \leq \sum_{k=1}^{k_0-1} \mathrm{Reg}(\mathcal{I}_k) + \sum_{k=k_0}^{k_1-1} \mathrm{Reg}(\mathcal{I}_k) + \sum_{k=k_1}^{N} \mathrm{Reg}(\mathcal{I}_k).$$

We first handle regret from intervals before $k_0$, which is the simplest case. Observe that that $\sum_{i=1}^{k_0-1} |\mathcal{I}_k| \leq T_{m^\star}^{\min}$. Combined with event $A_1$, this implies that

$$\sum_{k=1}^{k_0-1} \mathrm{Reg}(\mathcal{I}_k) \leq 2T_{k_0} \cdot \max_{a \in \mathcal{A}, t \in [T]} |\ell_t(a)| \leq \tilde{O}\left(\frac{\tau^3}{\gamma^2} \cdot d_{m^\star} \log^{3/2}(2/\delta) + \tau \log^2(2/\delta) + \tau K \log^{1/2}(2/\delta)\right), \tag{19}$$

whenever $\kappa \leq 1/3$. For every other interval, we bound the regret as follows:
$$\begin{aligned}
\mathrm{Reg}(\mathcal{I}_k) &= \sum_{t \in \mathcal{I}_k} \ell_t(a_t) - \ell_t(\pi^\star(x_t)) \\
&= \sum_{t \in \mathcal{I}_k} \ell_t(a_t) - \ell_t(\pi_{m^\star}^{\mathrm{sq}}(x_t)) \quad \text{(using Proposition 20)} \\
&= \sum_{t \in \mathcal{I}_k} \ell_t(a_t) - \ell_t(\pi_{\widehat{m}_k}^{\mathrm{sq}}(x_t)) + \sum_{t \in \mathcal{I}_k} \ell_t(\pi_{\widehat{m}_k}^{\mathrm{sq}}(x_t)) - \ell_t(\pi_{m^\star}^{\mathrm{sq}}(x_t)).
\end{aligned}$$
For the first summation, using event $A_1$ and $A_4$, we have

$$\begin{aligned}
\sum_{t \in \mathcal{I}_k} \ell_t(a_t) - \ell_t(\pi_{\widehat{m}_k}^{\mathrm{sq}}(x_t)) &\leq \sum_{t \in \overline{\mathcal{I}}_k} \ell_t(a_t) - \ell_t(\pi_{\widehat{m}_k}^{\mathrm{sq}}(x_t)) + \tilde{O}\left(|\mathcal{I}_k \cap S_T| \cdot \tau\sqrt{\log(2/\delta)}\right) \\
&\leq \tilde{O}\left(\tau\sqrt{d_{\widehat{m}_k}|\mathcal{I}_k| \cdot K \log^2(2/\delta)}\right) + \tilde{O}\left(|\mathcal{I}_k \cap S_T| \cdot \tau\sqrt{\log(2/\delta)}\right).
\end{aligned}$$

The second summation is exactly zero when $k \geq k_1$, otherwise we invoke event $A_5$ and Lemma 22, which imply

$$\begin{aligned}
\sum_{t \in \mathcal{I}_k} \ell_t(\pi_{\widehat{m}_k}^{\mathrm{sq}}(x_t)) - \ell_t(\pi_{m^\star}^{\mathrm{sq}}(x_t)) &\leq |\mathcal{I}_k| \cdot (L(\pi_{\widehat{m}_k}^{\mathrm{sq}}) - L(\pi_{m_k^\star}^{\mathrm{sq}})) + \tilde{O}\left(\tau\sqrt{|\mathcal{I}_k|\log(2/\delta)}\right) \\
&\leq O\left(|\mathcal{I}_k|\sqrt{K \cdot \mathcal{E}_{\widehat{m}_k, m^\star}}\right) + \tilde{O}\left(\tau\sqrt{|\mathcal{I}_k|\log(2/\delta)}\right).
\end{aligned}$$

Combining these results, we get that

$$\begin{aligned}
&\sum_{k=k_0}^{k_1-1} \mathrm{Reg}(\mathcal{I}_k) + \sum_{k=k_1}^{N} \mathrm{Reg}(\mathcal{I}_k) \\
&\leq \tilde{O}\left(|S_T| \cdot \tau\sqrt{\log(2/\delta)}\right) + \tilde{O}\left(\sum_{k=k_1}^{N} \tau\sqrt{d_{\widehat{m}_k}|\mathcal{I}_k| \cdot K \log^2(2/\delta)}\right) \\
&\quad + \tilde{O}\left(\sum_{k=k_0}^{k_1-1} \tau\sqrt{d_{\widehat{m}_k}|\mathcal{I}_k| \cdot K \log^2(2/\delta)} + |\mathcal{I}_k|\sqrt{K \cdot \mathcal{E}_{\widehat{m}_k, m^\star}} + \tau\sqrt{|\mathcal{I}_k|\log(2/\delta)}\right). \tag{20}
\end{aligned}$$

From here we split into two cases.

**Regret after $k_1$** We first claim that if $\widehat{m}_k \geq m^\star$, it must be the case that $k = N$, or in other words, if it happens that we switch to a policy class containing $\pi^\star$, we never leave this class. Indeed, suppose that $\widehat{m}_k \geq m^\star$, and that at time $T_{k+1}$ we switched to $\widehat{m}_{k+1} \neq \widehat{m}_k$. Then it must have been the case that for $t = T_{k+1} - 1$, there was some $i$ for which

$$\widehat{\mathcal{E}}_{\widehat{m},i}(t) \geq 2\alpha_{i,t}.$$

But since we have $t \geq T_i^{\min}$, event $A_3$ then implies that

$$\frac{3}{2}\mathcal{E}_{\widehat{m},i} \geq \widehat{\mathcal{E}}_{\widehat{m},i}(t) - \alpha_{i,t} \geq \alpha_{i,t} > 0,$$

which is a contradiction because $\mathcal{E}_{i,j} = 0$ for all $m^\star \leq i < j$ by Proposition 20. We conclude that

$$\sum_{k=k_1}^{N} \tau\sqrt{d_{\widehat{m}_k}|\mathcal{I}_k| \cdot K \log^2(2/\delta)} = \tau\sqrt{d_{\widehat{m}_N}|\mathcal{I}_N| \cdot K \log^2(2/\delta)},$$

in the case where $k_1 = N$, and is zero otherwise. It remains to show that if we happen to overshoot the class $m^\star$ (i.e., $\widehat{m}_N > m^\star$), then $d_{\widehat{m}_N}$ is not too large relative to $d_{m^\star}$.

Suppose $\widehat{m}_N > m^\star$, let $j := \widehat{m}_{k_1}$ and consider the epoch prior to $N$. At the time $t = T_N - 1$ at which we switched, the definition of $\widehat{m}$ implies that we must have had

$$\widehat{\mathcal{E}}_{\widehat{m},m^\star}(t) \leq 2\alpha_{m^\star,t} \quad \text{and} \quad \widehat{\mathcal{E}}_{\widehat{m},j}(t) \geq 2\alpha_{j,t}.$$

Using event $A_3$, this implies that $\mathcal{E}_{\widehat{m},m^\star} \leq 6\alpha_{m^\star,t}$ and $\frac{3}{2}\mathcal{E}_{\widehat{m},j} \geq \alpha_{j,t}$. However, Proposition 20 implies $\mathcal{E}_{\widehat{m},j} = \mathcal{E}_{\widehat{m},m^\star}$, and so we get that $\alpha_{j,t} \leq 9\alpha_{m^\star,t}$. Expanding out the definition for the $\alpha_{m,t}$, and defining $c_1 = \frac{\tau^6}{\gamma^4} \cdot \frac{1}{K^\kappa t^{1-\kappa}}$ and $c_2 = \frac{\tau^{10}}{\gamma^8} \cdot \frac{\log(2/\delta_0)}{t}$, this implies

$$C_1 \cdot \left(c_1 \cdot d_j^{1/2} \log^2(2d_j/\delta_0) + c_2 \cdot d_j\right) \leq 9C_1 \cdot \left(c_1 \cdot d_{m^\star}^{1/2} \log^2(2d_{m^\star}/\delta_0) + c_2 \cdot d_{m^\star}\right)$$

or, simplifying,

$$c_1 \cdot d_j^{1/2} \log^2(2d_j/\delta_0) \leq 9c_1 \cdot d_{m^\star}^{1/2} \log^2(2d_{m^\star}/\delta_0) + c_2(9d_{m^\star} - d_j)$$

Now consider two cases. If $d_j < 9d_{m^\star}$ we are done. If not, the inequality above implies $d_j^{1/2} \log^2(2d_j/\delta_0) \leq d_{m^\star}^{1/2} \log^2(2d_{m^\star}/\delta_0)$. We conclude that $d_j = O(d_{m^\star})$, so

$$\sum_{k=k_1}^{N} \tau\sqrt{d_{\widehat{m}_k}|\mathcal{I}_k| \cdot K \log^2(2/\delta)} = O\left(\tau\sqrt{d_{m^\star}|\mathcal{I}_N| \cdot K \log^2(2/\delta)}\right). \tag{21}$$

**Regret between $k_0$ and $k_1$** Let $k_0 \leq k < k_1$. We will bound the term

$$\tau\sqrt{d_{\widehat{m}_k}|\mathcal{I}_k| \cdot K \log^2(2/\delta)} + |\mathcal{I}_k|\sqrt{K \cdot \mathcal{E}_{\widehat{m}_k,m^\star}}$$

appearing in (20). For the first term, note that we trivially have $d_{\widehat{m}_k} \leq d_{m^\star}$. For the second, consider $t = T_{k+1} - 2$. Since we did not switch at this time, we have $\widehat{\mathcal{E}}_{\widehat{m}_k,m^\star}(t) \leq 2\alpha_{m^\star,t}$. Combined with event $A_3$, since $t \geq T_{m^\star}^{\min} - 1$, this implies that $\mathcal{E}_{\widehat{m}_k,m^\star} \leq 6\alpha_{m^\star,t}$, and so

$$|\mathcal{I}_k|\sqrt{K \cdot \mathcal{E}_{\widehat{m}_k,m^\star}} \leq \tilde{O}\left(\frac{\tau^3}{\gamma^2} \cdot |\mathcal{I}_k|\frac{K^{\frac{1}{2}(1-\kappa)}d_{m^\star}^{1/4}\log(2/\delta)}{T_{k+1}^{\frac{1}{2}(1-\kappa)}}\right) + \tilde{O}\left(\frac{\tau^5}{\gamma^4} \cdot |\mathcal{I}_k|\sqrt{\frac{Kd_{m^\star}\log(2/\delta)}{T_{k+1}}}\right)$$

$$\leq \tilde{O}\left(\frac{\tau^3}{\gamma^2} \cdot K^{\frac{1}{2}(1-\kappa)}|\mathcal{I}_k|^{\frac{1}{2}(1+\kappa)}d_{m^\star}^{1/4}\log(2/\delta)\right) + \tilde{O}\left(\frac{\tau^5}{\gamma^4} \cdot \sqrt{|\mathcal{I}_k|Kd_{m^\star}\log(2/\delta)}\right). \tag{22}$$

**Final result** We combine equations (19), (20), (21), and (22), and use the bound on $|S_T|$ from event $A_2$ to get

$$\text{Reg} \leq \tilde{O}\left(\sum_{k=1}^{N} \tau\sqrt{K|\mathcal{I}_k|d_{m^\star}\log^2(2/\delta)} + \tau\sqrt{\log(2/\delta)}K^\kappa T^{1-\kappa} + \frac{\tau^3}{\gamma^2} \cdot d_{m^\star}\log^{3/2}(2/\delta)\right)$$

$$+ \quad \tilde{O}\left(\sum_{k=1}^{N}\frac{\tau^3}{\gamma^2} \cdot K^{\frac{1}{2}(1-\kappa)}|\mathcal{I}_k|^{\frac{1}{2}(1+\kappa)}d_{m^\star}^{1/4}\log(2/\delta) + \sum_{k=1}^{N}\frac{\tau^5}{\gamma^4} \cdot \sqrt{K|\mathcal{I}_k|d_{m^\star}\log(2/\delta)}\right).$$

Using that $\tau/\gamma \geq 1$ and Jensen's inequality, this simplifies to an upper bound of

$$\leq \tilde{O}\left(\tau\sqrt{\log(2/\delta)}K^{\kappa}T^{1-\kappa} + \frac{\tau^3}{\gamma^2} \cdot K^{\frac{1}{2}(1-\kappa)}(Tm^{\star})^{\frac{1}{2}(1+\kappa)}d_{m^{\star}}^{1/4}\log(2/\delta) + \frac{\tau^5}{\gamma^4} \cdot \sqrt{KTm^{\star}d_{m^{\star}}\log^2(2/\delta)}\right).$$

For the choice $\kappa = 1/3$, this becomes

$$\text{Reg} \leq \tilde{O}\left(\tau\sqrt{\log(2/\delta)}K^{1/3}T^{2/3} + \frac{\tau^3}{\gamma^2} \cdot K^{1/3}(Tm^{\star})^{2/3}d_{m^{\star}}^{1/4}\log(2/\delta) + \frac{\tau^5}{\gamma^4} \cdot \sqrt{KTm^{\star}d_{m^{\star}}\log^2(2/\delta)}\right)$$

$$\leq \tilde{O}\left(\frac{\tau^3}{\gamma^2} \cdot (Kd_{m^{\star}})^{1/3}(Tm^{\star})^{2/3}\log(2/\delta) + \frac{\tau^5}{\gamma^4} \cdot \sqrt{KTm^{\star}d_{m^{\star}}\log^2(2/\delta)}\right).$$

Whenever this regret bound is non-trivial, both terms above can be upper bounded as

$$\text{Reg} \leq \tilde{O}\left(\frac{\tau^4}{\gamma^3} \cdot (Kd_{m^{\star}})^{1/3}(Tm^{\star})^{2/3}\log(2/\delta)\right).$$

For the choice $\kappa = 1/4$, we have

$$\text{Reg} \leq \tilde{O}\left(\tau\sqrt{\log(2/\delta)}K^{1/4}T^{3/4} + \frac{\tau^3}{\gamma^2} \cdot K^{3/8}(Tm^{\star})^{5/8}d_{m^{\star}}^{1/4}\log(2/\delta) + \frac{\tau^5}{\gamma^4} \cdot \sqrt{KTm^{\star}d_{m^{\star}}\log^2(2/\delta)}\right).$$

To simplify the middle term above we consider two cases. If $K^{1/8}d_{m^{\star}}^{1/4} \leq (Tm^{\star})^{1/8}$, then $K^{3/8}(Tm^{\star})^{5/8}d_{m^{\star}}^{1/4} \leq K^{1/4}(Tm^{\star})^{3/4}$. If this does not hold, then we have

$$K^{3/8}(Tm^{\star})^{5/8}d_{m^{\star}}^{1/4} = K^{3/8}\sqrt{Tm^{\star}}d_{m^{\star}}^{1/4} \cdot (Tm^{\star})^{1/8} \leq \sqrt{KTm^{\star}d_{m^{\star}}}.$$

Combining these results and using again that $\tau/\gamma \geq 1$, we have

$$\text{Reg} \leq \tilde{O}\left(\frac{\tau^3}{\gamma^2} \cdot K^{1/4}(Tm^{\star})^{3/4}\log(2/\delta) + \frac{\tau^5}{\gamma^4} \cdot \sqrt{KTm^{\star}d_{m^{\star}}}\log(2/\delta)\right). \qquad \square$$

### D.3 Proofs for remaining results

**Proof of Theorem 4.** This result is a fairly immediate consequence of Theorem 3. Let $i \geq 1$ be such that $e^{i-1} \leq d_{m^{\star}} \leq e^i$. Then the feature map $\bar{\phi}^i$ must not have been removed by the duplicate removal step. Moreover, since $\bar{\phi}^i$ was chosen to be the largest feature map with dimension bounded by $e^i$, the policy class it induces must contain the policy class induced by $\phi^{m^{\star}}$ by nestedness, and the realizability assumption is preserved by the new set of feature maps. Lastly, we observe that $i \leq \log(d_{m^{\star}}) \leq \log(T)$, and that $d_i \leq e \cdot d_{m^{\star}}$, leading to the result. $\qquad \square$

## Footnotes

[8]Code is publicly available at `https://github.com/akshaykr/oracle_cb`.

[9]Note that $\Sigma_1 > 0$.