[Reviews · NeurIPS 2019]

Reviewer 1



The authors provide a clear descriptions of their setting and methods. They clearly point out where they can make improvements and how this setting is novel. They introduce a new structured bandit problem that is interesting and could have a significant impact on the field. Line 163, it would be more easy to read if you change \Delta_{\hat{m}, i} to \Delta_{\hat{m}, j} as this would reflect the naming structure from your previous part of the paper.

Reviewer 2



This paper proposes a novel problem setting about the model specification in the multi-armed bandit problem. The method proposed by authors are quite persuasive, and the theoretical result consistent with the expected theoretical ideal result predicted from previous researches of the linear bandit. However, I have two concerns. First, I could not understand why authors dervise their algorithm from a method proposed in the research of adversarial bandit problem. Second, it is not necessary, but authors are recommended to show their experimental results. I think that it is not so difficult to experiment with their problem setting. Originality: 5

Reviewer 3



Overall, this paper presents a novel algorithm for a challenging problem with solid theoretical analysis. The theorems are well supported with theoretical proofs. The results are promising. The paper is well written and clearly organized. All contributions are well presented and explained in the paper. And I like the discussion provided in this about the intuition behind the algorithm and some remaining questions. The discussion helps readers to understand the main idea better.

[Author Response · NeurIPS 2019]

We thank the reviewers for their valuable comments and time. Please see responses to individual questions below.

**Why Exp4 or adversarial bandits algorithm? (R1 and R2)**

Currently, we use Exp4 as a sub-algorithm within our new algorithm, LinCB.MS. The exact choice of sub-algorithm is not important, except that it must provide an *agnostic* regret guarantee. Here, "agnostic" means that the algorithm provides an $O(\sqrt{T})$ regret guarantee against the best policy in class $\Pi$, regardless of whether or not the loss distribution is *realizable* in the sense of Eq. (1). A number of other stochastic CB algorithms enjoy this property, including PolicyElimination [2] and ILOVETOCONBANDITS [1]; these algorithms could be used in place of Exp4. However, LinUCB is *not* an agnostic algorithm and hence is not a valid choice. Among the agnostic algorithms, we chose EXP4 because it is simple to describe and familiar to readers, even though it was originally designed for the adversarial setting.

The reason the agnostic guarantee is required is that LinCB.MS may invoke the sub-algorithm with a policy class $\Pi_m$ that is too small to contain the true parameter $\beta^\star$. It is important for our analysis that the sub-algorithm have low regret against $\Pi_m$ during this time, even though $\Pi_m$ doesn't satisfy the realizability assumption (1). Unfortunately, LinUCB does not enjoy a low regret in the absence of realizability, so we cannot use it here.

This is mentioned briefly at the beginning of section 3 on page 4 of the submission, but we are happy to expand the discussion. We also remark that using stochastic CB algorithms that better adapt to the distribution structure (e.g., to achieve instance-dependent guarantees) is a nice direction for future research.

**Validation experiments (R2 and R3)**

We believe that our paper represents a substantial theoretical contribution and stands on its own merits even without experiments. Nonetheless, we have performed some basic validation experiments, and the initial results are quite nice. Thank you for encouraging us to try this!

We built our experiments on top of an open source implementation of LinUCB and ILOVETOCONBANDITS which has previously been used in a number of experimental works on contextual bandits [4, 3, 5]. For computational efficiency, our implementation of LinCB.MS uses ILOVETOCONBANDITS [1] as the base learner instead of EXP4, which, as discussed above, suffices for our theoretical guarantees. In Figure 1, we evaluate three algorithms (LinUCB, our algorithm LinCB.MS, and ILOVETOCONBANDITS with knowledge of $d_{m^\star}$, which we call Oracle) on a simple synthetic problem with $d_{m^\star} = 10$ and ambient dimension $d = 1000$. We perform 20 replicates and tune hyperparameters for each algorithm, visualizing the cumulative regret, averaged over replicates. We see that LinCB.MS consistently outperforms LinUCB, and sometimes even outperforms Oracle. This

Figure 1: Validation experiments

latter phenomenon can be explained by the fact that while LinCB.MS typically advances to $d_m > d_{m^\star}$ (typically 32 dimensions), it sometimes stays below $d_{m^\star}$ (e.g., 8 dimensions), where it can learn a near optimal policy faster than Oracle.

We will definitely add these and related experiments to the final version of the paper, and provide a detailed description of our experimental methodology.

# References

[1] Alekh Agarwal, Daniel Hsu, Satyen Kale, John Langford, Lihon Li, and Robert E. Schapire. Taming the monster: A fast and simple algorithm for contextual bandits. In *International Conference on Machine Learning*, 2014.

[2] Miroslav Dudik, Daniel Hsu, Satyen Kale, Nikos Karampatziakis, John Langford, Lev Reyzin, and Tong Zhang. Efficient optimal learning for contextual bandits. In *Conference on Uncertainty in Artificial Intelligence*. AUAI Press, 2011.

[3] Dylan J. Foster, Alekh Agarwal, Miroslav Dudik, Haipeng Luo, and Robert Schapire. Practical contextual bandits with regression oracles. In *International Conference on Machine Learning*, 2018.

[4] Akshay Krishnamurthy, Alekh Agarwal, and Miro Dudik. Contextual semibandits via supervised learning oracles. In *Advances In Neural Information Processing Systems*, pages 2388–2396, 2016.

[5] Akshay Krishnamurthy, Zhiwei Steven Wu, and Vasilis Syrgkanis. Semiparametric contextual bandits. In *International Conference on Machine Learning*, pages 2781–2790, 2018.


[Meta-Review · NeurIPS 2019]

This paper formulated and proposed a solution to the problem of model selection in contextual bandits. All of the reviewers pointed out the importance of this problem and the novelty of the authors' solution. Overall, the reviews were positive about the paper and their concerns were addressed in the authors' response. The authors should make sure to update their submission to include the reviewers' suggestions, especially the empirical evaluation done in the author response.